# Molecular mechanism of decision-making in glycosaminoglycan biosynthesis

Douglas Sammon [1], Anja Krueger [1], Marta Busse-Wicher [1,5], Rhodri Marc Morgan [1,6], Stuart M. Haslam [1], Benjamin Schumann [2,3], David C. Briggs [1,4] ✉ & Erhard Hohenester [1] ✉

Two major glycosaminoglycan types, heparan sulfate (HS) and chondroitin sulfate (CS), control many aspects of development and physiology in a type-specific manner. HS and CS are attached to core proteins via a common linker tetrasaccharide, but differ in their polymer backbones. How core proteins are specifically modified with HS or CS has been an enduring mystery. By reconstituting glycosaminoglycan biosynthesis in vitro, we establish that the CS-initiating *N*-acetylgalactosaminyltransferase CSGALNACT2 modifies all glycopeptide substrates equally, whereas the HS-initiating *N*-acetylglucosaminyltransferase EXTL3 is selective. Structure-function analysis reveals that acidic residues in the glycopeptide substrate and a basic exosite in EXTL3 are critical for specifying HS biosynthesis. Linker phosphorylation by the xylose kinase FAM20B accelerates linker synthesis and initiation of both HS and CS, but has no effect on the subsequent polymerisation of the backbone. Our results demonstrate that modification with CS occurs by default and must be overridden by EXTL3 to produce HS.

Heparan sulfate (HS) and chondroitin sulfate (CS) are protein-attached *O*-linked glycosaminoglycan (GAG) chains that play critical roles in animal development and physiology. They control the diffusion and cellular signalling of many morphogens, growth factors and cytokines; they are essential for the structural integrity of extracellular matrix; and they are intimately involved in many cellular processes such as adhesion, migration, and endocytosis[1–3]. Genetic disruption of GAG biosynthesis in mice results in embryonic lethality at the eight-cell stage (deletion of both HS and CS)[4] or at gastrulation (deletion of HS only)[5]. In humans, malfunction of GAG biosynthesis underlies a range of severe disorders, frequently affecting skeletal development and cognitive functions[6].

HS and CS are covalently attached to a set of approximately 40 core proteins, forming the major proteoglycans (PGs) of the basement membrane (perlecan, agrin, collagen XVIII), cartilage (aggrecan), and the cell surface (syndecans, glypicans)[7]. PGs may be modified exclusively with HS or CS chains or with a mixture of both. The identity of the GAG chain is important for many PG functions: in the nervous system, for instance, HS is an attractive signal through receptor protein tyrosine phosphatase σ, whereas CS is repellent[8]. How the appropriate GAG chains are attached to a given PG core protein is a key question.

HS and CS are assembled by a series of glycosyltransferases (GTs) within the Golgi apparatus (Fig. 1a)[9–11]. The first steps are common to both GAG types. First, one of two closely related xylosyltransferases, XT1 or XT2, adds Xyl to Ser-Gly (rarely Ser-Ala) sequons in unstructured regions of the core protein[12]. Next, the galactosyltransferase B4GALT7 adds Gal and the kinase FAM20B phosphorylates Xyl to form Gal-Xyl-2-phosphate (Gal-Xyl2P)[13,14]. Then, another Gal is added to the phosphorylated disaccharide by the galactosyltransferase B3GALT6. Finally, GlcA is added by the glucuronyltransferase B3GAT3 to complete the linker tetrasaccharide common to HS and

[1]Department of Life Sciences, Imperial College London, London SW7 2AZ, UK. [2]Department of Chemistry, Imperial College London, London W12 0BZ, UK. [3]Chemical Glycobiology Laboratory, The Francis Crick Institute, London NW1 1AT, UK. [4]Signalling and Structural Biology Laboratory, The Francis Crick Institute, London NW1 1AT, UK. [5]Present address: Abzena, Babraham Research Campus, Cambridge CB22 3AT, UK. [6]Present address: ZoBio, 2333 CH Leiden, Netherlands. ✉e-mail: david.briggs@crick.ac.uk; e.hohenester@imperial.ac.uk

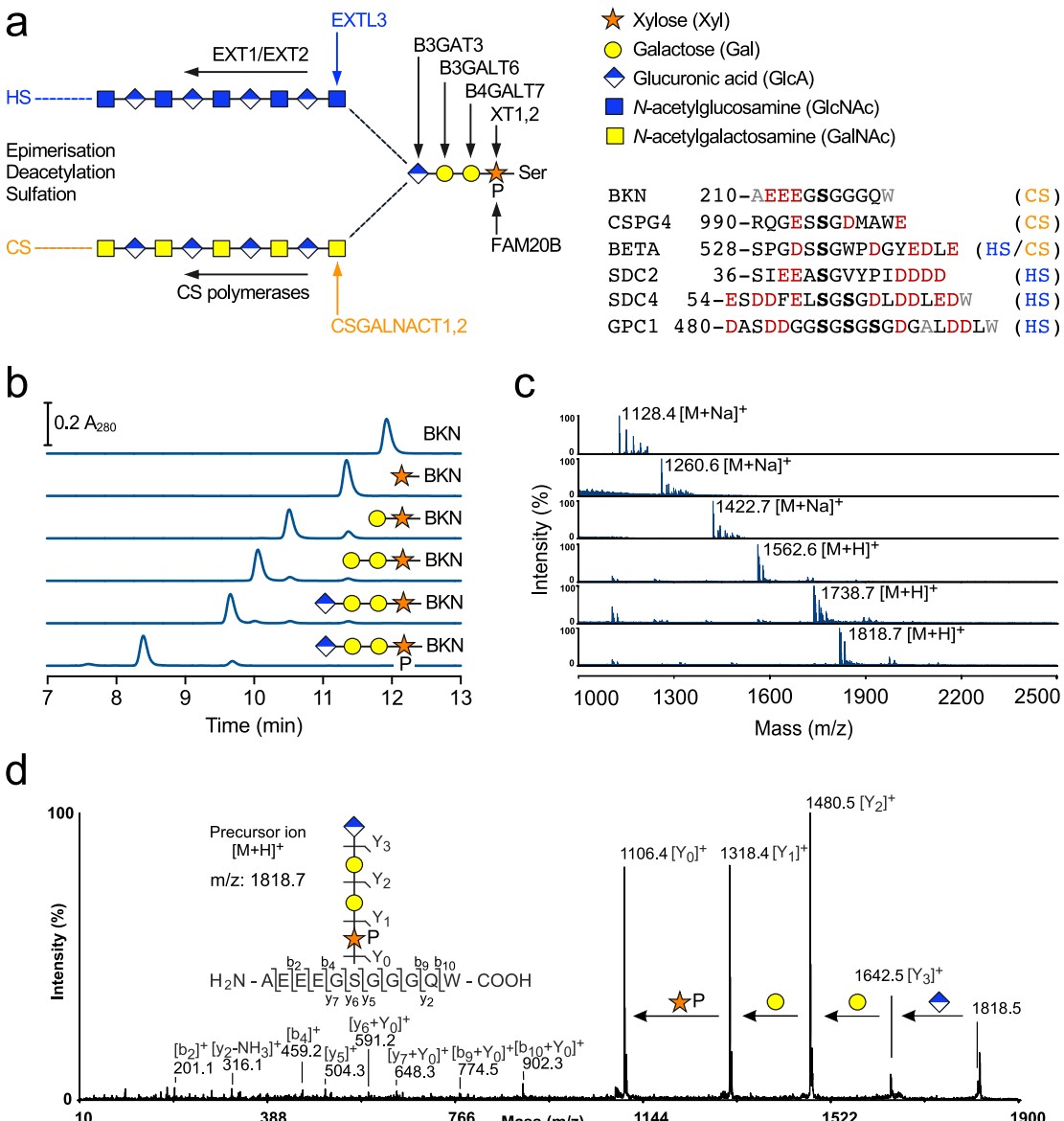

**Fig. 1 | One-pot multienzyme synthesis of peptides with linker tetrasaccharides.** **a** Schematic representation of the GAG biosynthetic pathway, with all relevant enzymes indicated. Sugar symbols are defined on the right. P indicates phosphorylation of the Xyl 2-OH group. Also shown are the sequences of the peptides used in this study. The parent proteins are: BKN, human bikunin; CSPG4, human chondroitin sulfate proteoglycan 4; BETA, mouse transforming growth factor β receptor 3/betaglycan; SDC2, human syndecan-2; SDC4, human syndecan-4, GPC1, human glypican-1. Modified serines are in bold, acidic residues are in red and residues altered for detection or stability are in grey. **b** High-performance liquid chromatography analysis of one-pot multienzyme reaction products obtained with the BKN peptide. **c** Matrix-assisted laser desorption/ionisation time-of-flight mass spectrometry (MS) of peak fractions. **d** Tandem MS analysis of the TetraP-BKN glycopeptide. For the synthesis of other glycopeptides, see Supplementary Fig. 4. Source data are provided as a Source Data file.

CS: GlcAβ1-3Galβ1-3Galβ1-4Xyl2P-β-Ser. Sulfation of the linker has been observed in CS[10], but the functional significance of this modification is unclear: overexpression in CHO cells of the linker-modifying 6-O-sulfotransferase had no effect on the global HS/CS ratio[11].

The completed linker tetrasaccharide represents the point of bifurcation in GAG biosynthesis: either EXTL3 adds an α-linked GlcNAc and initiates HS polymerisation or one of two closely related CSGAL-NACTs adds a β-linked GalNAc and initiates CS polymerisation[9–11]. The respective pentasaccharides (referred to as primed linkers in this study) are elongated by heterodimeric bifunctional polymerases to form the regular backbones of HS and CS. The [-4GlcAβ1-4GlcNAcα1-]$_n$ repeats of HS are synthesised by EXT1/EXT2; this process is now well-understood thanks to recent structure determinations[15,16]. The [-4GlcAβ1-3GalNAcβ1-]$_n$ repeats of CS are synthesised by polymerases assembled from four related subunits: CHSY1, CHSY3, CHPF and CHPF2; different subunit pairings synthesise CS chains of different lengths, but how this is achieved is unclear[17,18]. During the polymerisation of the GAG chains, the disaccharide repeats are modified by deacetylation of GlcNAc followed by N-sulfation (HS only), O-sulfation at various positions and epimerisation of GlcA to iduronic acid (epimerised CS is called dermatan sulfate, DS)[9,10]. In addition to imparting GAGs with a large net negative charge, these modifications have been shown to encode specific functions, thereby vastly expanding the functional repertoire of the ≈40 core proteins[1,19].

What determines whether a GAG chain becomes HS or CS? GAG biosynthesis is not templated and, therefore, must depend on the specificity and/or subcellular localisation of the biosynthetic machinery to achieve fidelity. A series of elegant experiments in the 1990s showed that the sequence context of the modified Ser-Gly sequons is important for HS/CS selection[20,21], but because these experiments were

done in cells, it was not possible to relate the biosynthetic outcome to specific steps in the pathway. A similar limitation applies to another study that showed preferential HS modification of glypican-1 in cells[22].

We decided to tackle the question of HS/CS selection from the bottom up by studying GAG biosynthesis in a fully reconstituted system, using soluble enzymes and defined peptide and protein substrates, as well as structural analysis. We show that selection of the GAG chain type occurs at the priming step: CSGALNACTs are capable of initiating CS synthesis on all GAG attachment sites, whereas EXTL3 requires specific features in the core protein to initiate HS synthesis. For sites modified with HS in vivo, the kinetic parameters favour priming by EXTL3. For all other sites, EXTL3 activity is negligible, resulting in modification with CS.

## Results

### One-pot multienzyme synthesis of glycopeptide substrates

An attractive hypothesis is that the local protein sequence of the GAG attachment site specifies the GAG type by interacting directly with one or both of the respective priming enzymes: EXTL3 for HS and CSGALNACTs for CS[23]. In order to test this hypothesis, we reconstituted the GAG biosynthetic pathway in vitro using soluble enzymes. GAG biosynthesis in vivo is carried out by type II transmembrane proteins whose catalytic domains are located within the Golgi lumen. We obtained secreted soluble proteins by replacing the cytoplasmic and transmembrane regions with a signal peptide[24]. In some cases, secretion required fusion to maltose-binding protein (MBP) or co-expression of two enzyme subunits. In total, we produced eight single-chain proteins (XT1, B4GALT7, MBP-B3GALT6, B3GAT3, FAM20B, EXTL3, MBP-CSGALNACT1 and CSGALNACT2) and two obligate heterodimers (EXT1/EXT2 and CHSY3/CHPF) (Supplementary Fig. 1, Supplementary Table 1). This panel of recombinant enzymes allowed us to reconstitute GAG biosynthesis up to and including the polymerisation of the HS and CS backbone (Fig. 1a).

GAG attachment sites are invariably located within unstructured regions of the core proteins (Supplementary Fig. 2), likely because the core proteins are already fully folded in the Golgi compartment, and the xylosyltransferases need to gain access to several amino acid residues on either side of the Ser-Gly sequon[12]. This feature makes peptides excellent substrates for enzymological studies. We selected a range of peptide substrates, including sites that in vivo are modified with CS (bikunin, BKN; CS proteoglycan 4, CSPG4), a site that in vivo is modified with a mixture of HS and CS (transforming growth factor β receptor 3/betaglycan, BETA), and sites that in vivo are modified predominantly with HS (syndecan-2, SDC2; syndecan-4, SDC4; glypican-1, GPC1)[7,25] (Fig. 1a). The SDC4 and GPC1 sequences contain, respectively, two and three Ser-Gly sequons in tandem.

We initially used the BKN peptide as the acceptor substrate to build GAG biosynthetic intermediates in vitro. Remarkably, co-incubation of the peptide with all the GAG linker enzymes (XT1, B4GALT7, B3GALT6, B3GAT3 and FAM20B), their cognate UDP-sugars and ATP, in a one-pot multienzyme (OPME) reaction[26] resulted in the near-complete (≈90%) conversion of BKN to the phosphorylated tetrasaccharide peptide, GlcA-Gal-Gal-Xyl2P-BKN (TetraP-BKN) (Fig. 1b). Mass spectrometry (MS) confirmed the identity of the reaction product and the correct tetrasaccharide structure (Fig. 1c, d). All intermediates could be obtained in similar purity: a reaction with just XT1 gave Xyl-BKN; a reaction with XT1 and B4GALT7 gave Gal-Xyl-BKN; a reaction with XT1, B4GALT7 and B3GALT6 gave Gal-Gal-Xyl-BKN; and a reaction of all four GTs but without FAM20B gave the unphosphorylated linker glycopeptide, GlcA-Gal-Gal-Xyl-BKN (Tetra-BKN) (Fig. 1b).

To examine the effect of Xyl phosphorylation on linker synthesis, we produced Gal-Xyl2P-BKN and Gal-Gal-Xyl2P-BKN by OPME synthesis and used these glycopeptides as acceptors in kinetic experiments (Xyl2P-BKN was not studied because it is known that B4GALT7 does

not modify phosphorylated xylosides[27]). We found that substrate phosphorylation dramatically increased the catalytic efficiency of B3GALT6 (632-fold increase in $k_{cat}/K_M$) and modestly increased the efficiency of B3GAT3 (6.4-fold increase in $k_{cat}/K_M$) (Supplementary Fig. 3). Therefore, although not essential for GAG biosynthesis, phosphorylation of the Gal-Xyl-BKN intermediate promotes linker completion by B3GALT6 and B3GAT3. To rationalise the preference of B3GALT6 for phosphorylated acceptor substrates, we used a predicted B3GALT6 structure and information on substrate binding from a related enzyme, B3GNT2 (24.4% sequence identity). This analysis predicts that the phosphate group of Xyl2P is located next to a highly basic region of B3GALT6 (Supplementary Fig. 3d).

We successfully used OPME reactions to assemble complete linker tetrasaccharides, with and without Xyl phosphorylation, on all other peptides, including those containing multiple Ser-Gly sequons (Fig. 1a, Supplementary Fig. 4, Supplementary Table 2). Remarkably, the tandem sequons of SDC4 and GPC1 were converted almost quantitatively to give the fully modified glycopeptides, demonstrating the extraordinary potential of OPME synthesis.

### HS but not CS initiation is sequence-specific

We scaled up the OPME reactions to obtain milligramme quantities of Tetra and TetraP glycopeptides for enzymological experiments with EXTL3 (initiates HS) and CSGALNACT1 and 2 (initiate CS). For CS initiation, we mostly used CSGALNACT2 because it was easier to produce than CSGALNACT1; the two enzymes are 60% identical, and we found no differences in their substrate specificity (see below).

We initially compared a GAG attachment site modified with CS in vivo (BKN peptide) and a site modified with HS in vivo (BETA peptide). CSGALNACT2 converted ≈55% of Tetra-BKN into GalNAc-Tetra-BKN in an overnight reaction, whereas EXTL3 converted only ≈5% of Tetra-BKN into GlcNAc-Tetra-BKN (Fig. 2a). Xyl phosphorylation (TetraP-BKN substrate) resulted in 100% product formation in the reaction with CSGALNACT2, but the reaction with EXTL3 was still only ≈25% complete. In sharp contrast, Tetra-BETA and TetraP-BETA were converted quantitatively into products by both CSGALNACT2 and EXTL3 (Fig. 2a), suggesting that BETA contains specific sequence features that make it a good substrate for EXTL3. Similar results were obtained with two other glycopeptides: TetraP-CSPG4 (derived from a CSPG) was a good substrate only for CSGALNACT2, whereas TetraP-SDC2 (derived from an HSPG) was a good substrate for both CSGALNACT2 and EXTL3 (Supplementary Fig. 5a).

Kinetic analysis using the UDP-Glo assay revealed that Xyl phosphorylation increased the catalytic efficiency of both EXTL3 and CSGALNACT2 towards all glycopeptides tested (7-fold to 44-fold increase in $k_{cat}/K_M$ depending on the enzyme-substrate combination; Fig. 2b). CSGALNACT2 showed no peptide sequence preference, producing comparable kinetic profiles with all TetraP-peptide substrates (Fig. 2b). A similar behaviour was confirmed for CSGALNACT1 (Supplementary Fig. 5b). In sharp contrast to CSGALNACTs, EXTL3 showed a strong preference for peptide sequences derived from PGs modified with HS in vivo: TetraP-BETA, TetraP-SDC2, TetraP-SDC4 and TetraP-GPC1 (Fig. 2b). With these substrates, we observed saturable Michaelis-Menten behaviour and apparent $K_M$ values in the low μM range, at least 10 times lower than for any CSGALNACT2-substrate combination. The collective data demonstrate that EXTL3 activity, but not CSGALNACT2 activity, is dependent on the peptide sequence of the acceptor substrate.

A commonly suggested determinant of HS preference is the presence of multiple Ser-Gly sequons in close proximity[20]. Our analysis of SDC4 and GPC1 glycopeptides (containing two and three Ser-Gly sequons, respectively) showed that they were good substrates for both EXTL3 and CSGALNACT2 (Fig. 2b) and that multiple linkers were being primed by both enzymes (Supplementary Fig. 5a). Therefore, multiple Ser-Gly sites do not prevent CS initiation, but appear to contain

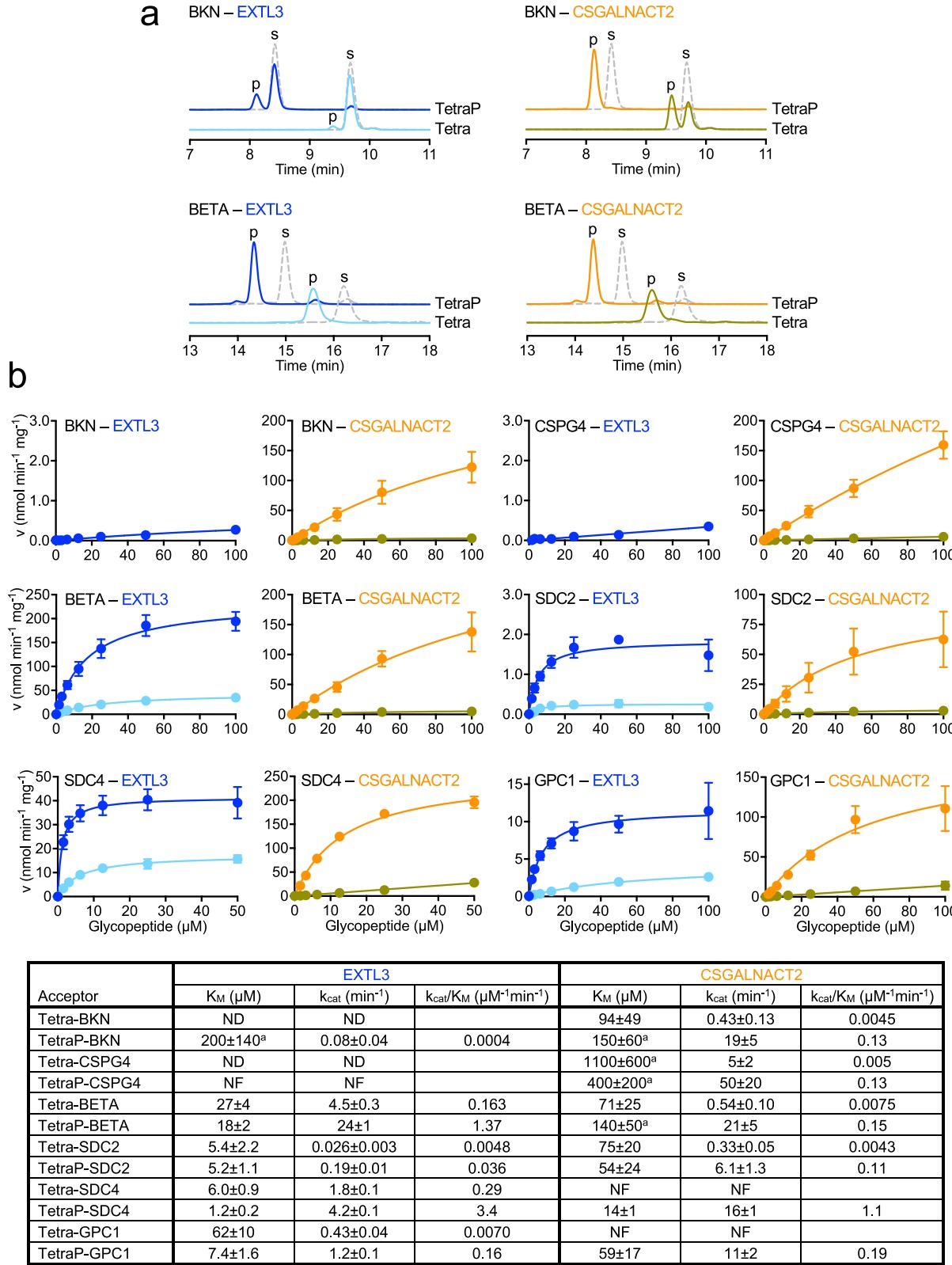

| | EXTL3 | | | CSGALNACT2 | | |
|---|---|---|---|---|---|---|
| Acceptor | $K_M$ (µM) | $k_{cat}$ (min$^{-1}$) | $k_{cat}/K_M$ (µM$^{-1}$min$^{-1}$) | $K_M$ (µM) | $k_{cat}$ (min$^{-1}$) | $k_{cat}/K_M$ (µM$^{-1}$min$^{-1}$) |
| Tetra-BKN | ND | ND | | 94±49 | 0.43±0.13 | 0.0045 |
| TetraP-BKN | 200±140[a] | 0.08±0.04 | 0.0004 | 150±60[a] | 19±5 | 0.13 |
| Tetra-CSPG4 | ND | ND | | 1100±600[a] | 5±2 | 0.005 |
| TetraP-CSPG4 | NF | NF | | 400±200[a] | 50±20 | 0.13 |
| Tetra-BETA | 27±4 | 4.5±0.3 | 0.163 | 71±25 | 0.54±0.10 | 0.0075 |
| TetraP-BETA | 18±2 | 24±1 | 1.37 | 140±50[a] | 21±5 | 0.15 |
| Tetra-SDC2 | 5.4±2.2 | 0.026±0.003 | 0.0048 | 75±20 | 0.33±0.05 | 0.0043 |
| TetraP-SDC2 | 5.2±1.1 | 0.19±0.01 | 0.036 | 54±24 | 6.1±1.3 | 0.11 |
| Tetra-SDC4 | 6.0±0.9 | 1.8±0.1 | 0.29 | NF | NF | |
| TetraP-SDC4 | 1.2±0.2 | 4.2±0.1 | 3.4 | 14±1 | 16±1 | 1.1 |
| Tetra-GPC1 | 62±10 | 0.43±0.04 | 0.0070 | NF | NF | |
| TetraP-GPC1 | 7.4±1.6 | 1.2±0.1 | 0.16 | 59±17 | 11±2 | 0.19 |

[a]$K_M$ exceeds the highest substrate concentration tested and is therefore not well determined by the data.

specific flanking sequences that make them good substrates for EXTL3. In order to study the effect of a tandem Ser-Gly site on HS initiation, we substituted the first serine in SDC4 with alanine (SDC4-A peptide). This variant with a single Ser-Gly sequon was still a good substrate for EXTL3, but the catalytic efficiency for TetraP-SDC4-A was 5.2-fold lower than for the parent glycopeptide with two modified Ser-Gly sequons (Supplementary Fig. 6).

## An exosite in EXTL3 determines the specificity

In cell-based experiments, Zhang and Esko (1994) showed that aromatic and acidic residues C-terminal of the Ser-Gly sequon in beta-glycan were important for HS modification[21]. To determine whether these residues were responsible for the EXTL3 peptide specificity observed above, we examined three variants of the BETA peptide: BETA-Y, in which the tryptophan in the +2 position was replaced with

**Fig. 2 | HS but not CS initiation is peptide sequence-dependent. a** High-performance liquid chromatography analysis of the priming reactions catalysed by EXTL3 and CSGALNACT2, using Tetra(P)-BKN and Tetra(P)-BETA as acceptors. Substrate (s) and product (p) peaks are labelled. The identity of the products was verified by mass spectrometry (Supplementary Table 2). The dashed grey lines represent the acceptor glycopeptides alone. **b** Kinetic analysis of the priming reactions catalysed by EXTL3 and CSGALNACT2, using the indicated glycopeptides as acceptors. The colour code is the same as in a: dark blue and orange for TetraP-

peptides, light blue and olive for Tetra-peptides. Initial rates were determined using the UDP-Glo assay over a range of glycopeptide concentrations in the presence of constant 100 μM UDP-sugar (UDP-GlcNAc for EXTL3, UDP-GalNAc for CSGAL-NACT2). Data points are shown as mean ± SEM (standard error of the mean) from n = 3 independent experiments and were fitted with the Michaelis–Menten equation. The kinetic parameters and their standard deviations from the non-linear fit are given in the table below the graphs. ND not determined, NF no fit obtained. Source data are provided as a Source Data file.

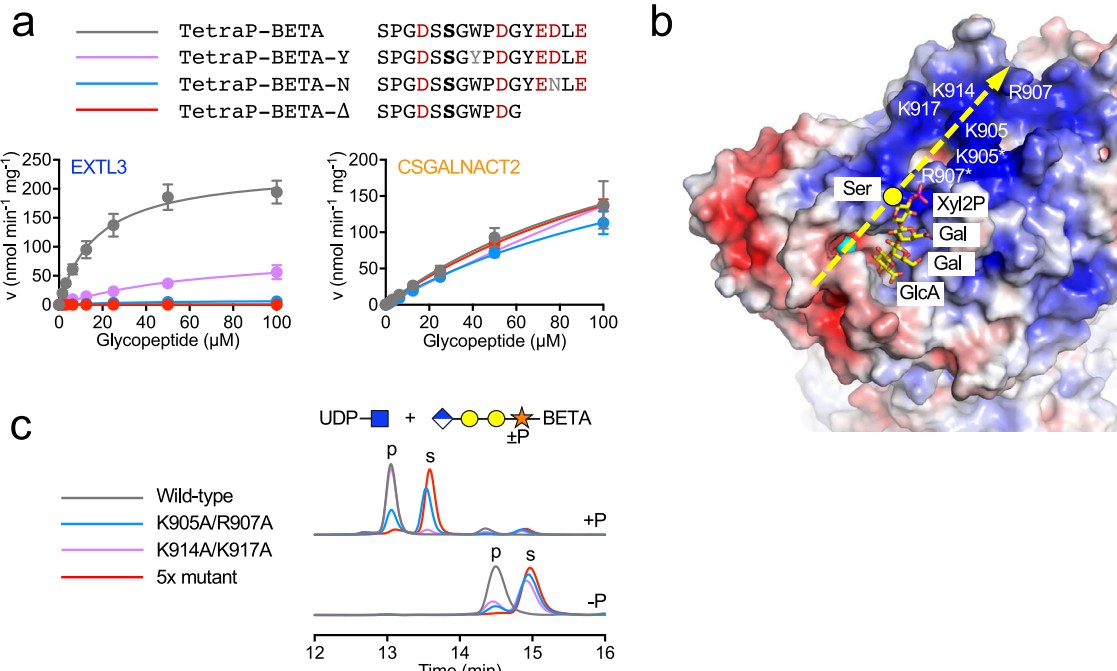

**Fig. 3 | EXTL3 peptide specificity is governed by a basic exosite. a** Kinetic analysis of the priming reactions catalysed by EXTL3 and CSGALNACT2, using the indicated TetraP-peptides as acceptors. Data points are shown as mean ± SEM from n = 3 independent experiments and were fitted with the Michaelis–Menten equation. **b** Electrostatic surface representation of the EXTL3 crystal structure in the vicinity of the GlcNAc transferase site (blue, positive potential; red, negative potential). The locations of selected basic residues are labelled. The $Mn^{2+}$ ion in the crystal structure is shown as a cyan sphere; UDP is obscured in this view. A phosphorylated

linker tetrasaccharide (shown in stick representation) was docked into the active site as described in "Methods". The yellow circle indicates the Cα atom of the modified serine. The predicted direction of the peptide backbone is indicated by the dashed yellow arrow. **c** High-performance liquid chromatography analysis of the priming reaction catalysed by wild-type and mutant EXTL3 using Tetra(P)-BETA as acceptors. Substrate (s) and product (p) peaks are labelled. Source data are provided as a Source Data file.

tyrosine; BETA-N, in which the aspartic acid in the +7 position was replaced with asparagine; and BETA-Δ, in which five residues were deleted from the C-terminus (Fig. 3a). TetraP linkers were assembled on these peptides using OPME synthesis. Overnight reactions (Supplementary Fig. 7a) and kinetic analysis (Fig. 3a) showed that substitution of a single acidic residue (BETA-N) drastically reduced priming by EXTL3 and that removal of three acidic residues (BETA-Δ) essentially abolished priming by EXTL3, whereas priming by CSGAL-NACT2 was unaffected in these variants. Substitution of tryptophan (BETA-Y) reduced $k_{cat}/K_M \approx 10$-fold, but had a less dramatic effect on EXTL3 priming than the substitution or deletion of acidic residues (Fig. 3a). Similar results were obtained with the SDC2 peptide: deletion of four aspartic acid residues from the C-terminus (SDC2-Δ peptide) reduced priming by EXTL3, but not by CSGALNACT2 (Supplementary Fig. 7a). Thus, acidic sequences C-terminal of the Ser-Gly sequon confer specificity for EXTL3.

In order to explain these data mechanistically, we determined crystal structures of an EXTL3 dimer lacking the coiled-coil (residues 154–919) in the apo form (1.58 Å resolution) and with UDP and $Mn^{2+}$ bound (2.10 Å resolution) (Supplementary Table 3). The latter structure was obtained by crystal soaking that also included TetraP-BETA,

but unfortunately the glycopeptide was excluded from the GlcNAc transferase site by a crystal lattice contact. The EXTL3 crystal structure closely matches a cryo-electron microscopy structure described previously[28] (root-mean-square deviation of 0.8 Å for 686 Cα atoms) but contains an additional 29 residues. Briefly, each subunit of the EXTL3 dimer contains an N-terminal GlcA transferase (GT47) domain that may be inactive[28] and a C-terminal GlcNAc transferase (GT64) domain. The large dimer interface features an intermolecular disulfide bond (Cys793–Cys915) that staples the C-terminal tail of one subunit to the body of the other subunit.

Because co-crystallisation of EXTL3 with glycopeptide substrates was not successful, we used information from a related EXTL2 structure[29] and the NMR structure of a linker tetrasaccharide[30] to construct a model of the EXTL3-acceptor substrate complex (see "Methods" for details). In this model, only the terminal GlcA is enclosed by the enzyme; all other linker sugars are surprisingly exposed (even the GlcA is less buried than in EXTL2 due to a Trp-to-Ser substitution in EXTL3). The Xyl2P phosphate group is ≈20 Å away from the GlcNAc transferase site and close to Arg907 (Fig. 3b). This arginine is part of a large basic surface region spanning the dimer interface that has been suggested to interact with the peptide portion of acceptor

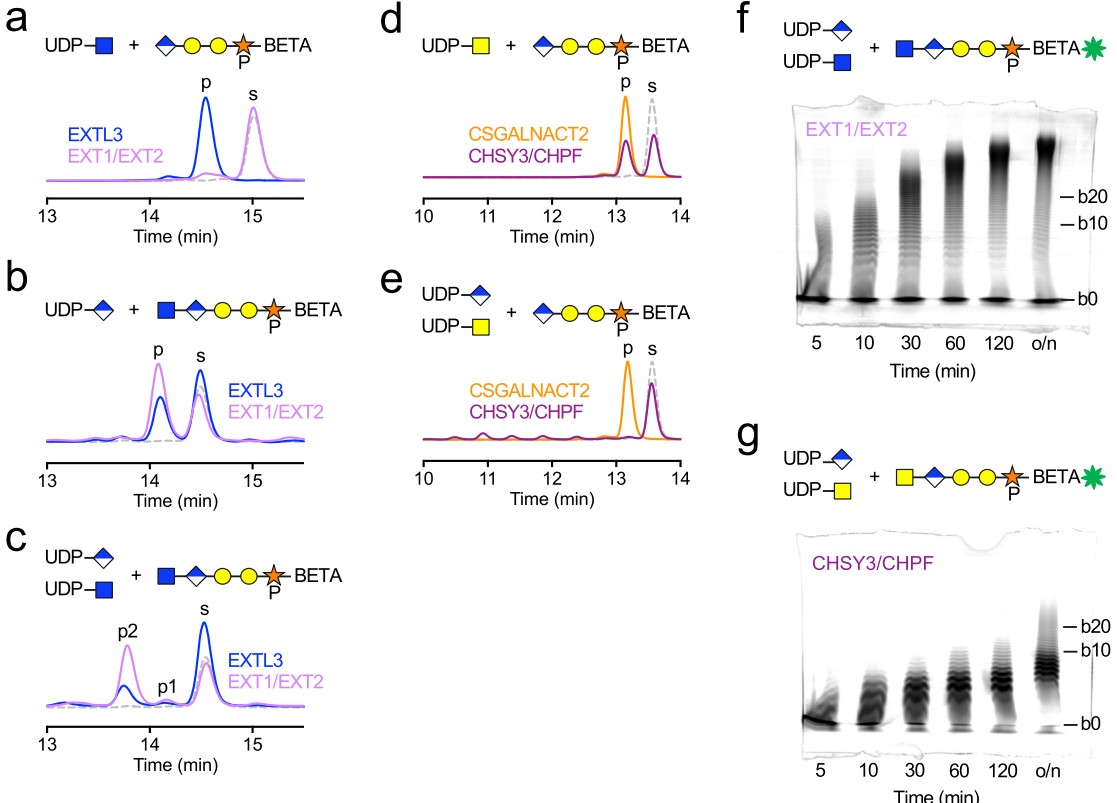

**Fig. 4 | Polymerisation of HS and CS backbones on peptides. a–c** High-performance liquid chromatography (HPLC) analysis of priming and elongation reactions catalysed by EXTL3 and EXT1/EXT2, using the indicated donors and acceptors. Substrate (s) and product (p) peaks are labelled. p1 is GlcA-GlcNAc-TetraP-BETA and p2 is GlcNAc-GlcA-GlcNAc-TetraP-BETA (verified by mass spectrometry; Supplementary Table 2). The dashed grey lines represent the acceptor glycopeptides alone. **d, e** HPLC analysis of priming and elongation reactions catalysed by CSGALNACT2 and CHSY3/CHPF, using the indicated donors and acceptors. **f** In vitro polymerisation of the HS backbone. GlcNAc-TetraP-BETA-5-FAM was incubated with EXT1/EXT2 and 100 equivalents each of UDP-GlcA and UDP-GlcNAc.

Reactions were stopped by boiling at the indicated time points (o/n, overnight). Reaction products were separated by SDS-PAGE and detected by in-gel fluorescence. The fastest migrating band (b0) corresponds to the unmodified glycopeptide. Each sugar addition results in an additional, slower migrating band; the 10th and 20th bands are labelled. **g** In vitro polymerisation of the CS backbone. GalNAc-TetraP-BETA-5-FAM was incubated with CHSY3/CHPF and 100 equivalents each of UDP-GlcA and UDP-GalNAc. The reaction products were analysed as in f. Representative gels from three independent experiments are shown. Source data are provided as a Source Data file.

substrates[28]. To determine whether basic residues are important for the selective modification of acidic acceptor glycopeptides, we mutated two pairs (K905A/R907A and K914A/K917A) and all five basic residues in the C-terminal tail of EXTL3 (5× mutant). All mutants assembled into disulfide-bonded dimers, suggesting that their folding was unaffected (Supplementary Fig. 7b). The activity of these EXTL3 mutants was assessed with Tetra-BETA and TetraP-BETA as acceptor substrates. Mutation of Arg905 and Arg907 reduced conversion of TetraP-BETA to product by ≈60%, and mutation of all five basic residues resulted in an almost complete loss of activity (Fig. 3c). These effects were even more pronounced with Tetra-BETA as substrate (Fig. 3c), indicating that both the phosphate and the acidic residues in TetraP-BETA contribute to EXTL3 binding.

To understand how CSGALNACT2 is capable of greater promiscuity than EXTL3, ColabFold[31] was used to predict the structure of CSGALNACT2. Light scattering analysis showed that, like EXTL3, CSGALNACT2 is a dimeric enzyme (Supplementary Fig. 8a). CSGALNACT2 was therefore predicted as a homodimer (Supplementary Fig. 8b), and a linker tetrasaccharide modelled into the GalNAc transferase site (see "Methods" for details). In this CSGALNACT2 model, the linker tetrasaccharide makes more extensive interactions with the enzyme than it does in EXTL3. Moreover, the protein surface surrounding the active site pocket in CSGALNACT2 has a balanced distribution of positive and negative charge, unlike in EXTL3 (Supplementary Fig. 8c). Thus, it seems that interactions with the

linker sugars rather than the peptide side chains dominate the recognition of acceptor substrates in CSGALNACT2.

## Polymerisation of HS and CS backbones in vitro

Having defined the rules governing HS/CS initiation, we next analysed the polymerisation of GAG chains. HS is polymerised by a heterodimer of EXT1 and EXT2[15,16]. CS polymerisation is more complex and less well understood. Four proteins have been described (CHSY1, CHSY3, CHPF and CHPF2) that pair in different combinations to produce an active polymerase[10]. We found that CS polymerase subunits could not be expressed alone; the only combination that produced sufficient soluble protein for enzymological analysis was CHSY3/CHPF (Supplementary Fig. 1b).

An open question is to what extent the polymerases are able to extend a linker tetrasaccharide without prior priming by EXTL3 or CSGALNACTs. EXT1/EXT2 showed negligible GlcNAc transferase activity with TetraP-BETA as a substrate, whereas EXTL3 efficiently converted this substrate into GlcNAc-TetraP-BETA (Fig. 4a). This result may explain why the deletion of EXTL3 in cells leads to a complete loss of HS synthesis[11,32]. Incubation of GlcNAc-TetraP-BETA with EXTL3 or EXT1/EXT2 and UDP-GlcA resulted in a single product peak, demonstrating that both enzymes are able to catalyse the addition of a sixth sugar, forming GlcA-GlcNAc-TetraP-BETA (Fig. 4b). Incubation of GlcNAc-TetraP-BETA with EXTL3 or EXT1/EXT2 and two equivalents each of UDP-GlcA and UDP-GlcNAc resulted in a second product

a

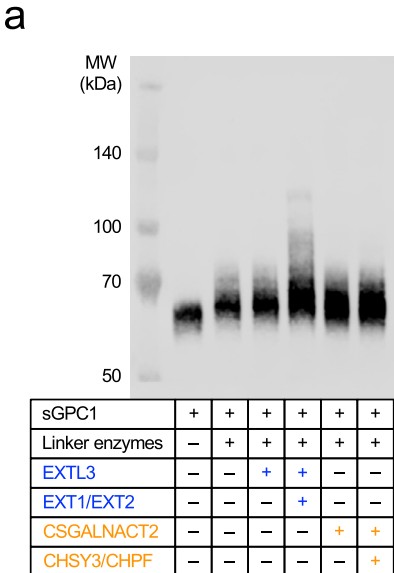

b

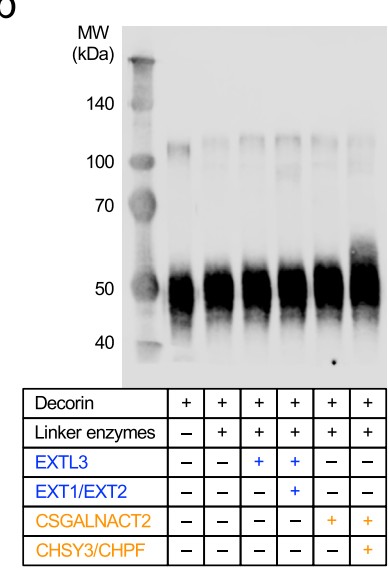

| sGPC1 | + | + | + | + | + | + |
|---|---|---|---|---|---|---|
| Linker enzymes | – | + | + | + | + | + |
| EXTL3 | – | – | + | + | – | – |
| EXT1/EXT2 | – | – | – | + | – | – |
| CSGALNACT2 | – | – | – | – | + | + |
| CHSY3/CHPF | – | – | – | – | – | + |

| Decorin | + | + | + | + | + | + |
|---|---|---|---|---|---|---|
| Linker enzymes | – | + | + | + | + | + |
| EXTL3 | – | – | + | + | – | – |
| EXT1/EXT2 | – | – | – | + | – | – |
| CSGALNACT2 | – | – | – | – | + | + |
| CHSY3/CHPF | – | – | – | – | – | + |

**Fig. 5 | Polymerisation of HS and CS backbones on folded core proteins. a** A soluble glypican-1 construct containing a C-terminal FLAG tag (sGPC1) was expressed in xylosyltransferase-deficient CHO cells. Purified sGPC1 was incubated for 1 h with the indicated biosynthetic enzymes (linker enzymes: XT1, B4GALT7, B3GALT6, B3GAT3 and FAM20B), their cognate UDP-sugars and ATP. Reaction products were boiled, separated by SDS-PAGE, transferred onto a nitrocellulose

membrane, and detected by an anti-FLAG antibody. Molecular weight markers are labelled. **b** FLAG-tagged decorin was expressed in xylosyltransferase-deficient CHO cells and analysed as described in a. The faint bands at ≈100 kDa likely are SDS-resistant decorin dimers. Representative Western blots from three independent experiments are shown. Source data are provided as a Source Data file.

peak (GlcNAc-GlcA-GlcNAc-TetraP-BETA), demonstrating that both enzymes have the potential to elongate the HS chain (Fig. 4c). However, EXT1/EXT2 seemed to be better at doing so than EXTL3. Further experiments showed that elongation by EXTL3 and EXT1/EXT2 was the same in the presence and absence of Xyl phosphorylation (Supplementary Fig. 9a–c). Therefore, after the initiation/priming step, Xyl phosphorylation no longer enhances HS biosynthesis.

CHSY3/CHPF was able to transfer GalNAc onto TetraP-BETA, albeit not as efficiently as CSGALNACT2 (Fig. 4d). This result may explain why deletion of both CSGALNACTs only reduces CS chain synthesis but does not abolish it altogether[11]. When CHSY3/CHPF was incubated with TetraP-BETA and two equivalents each of UDP-GalNAc and UDP-GlcA, several new peaks were identified by high-performance liquid chromatography, likely corresponding to a nascent CS polymer (Fig. 4e).

To study HS and CS chain elongation beyond the first few sugar additions, we established an in-gel fluorescence assay with a 5-FAM-labelled BETA peptide. We assembled primed tetrasaccharides on this peptide using OPME synthesis, with and without Xyl phosphorylation, giving GlcNAc-Tetra(P)-BETA-5-FAM and GalNAc-Tetra(P)-BETA-5-FAM. These glycopeptides were incubated, respectively, with EXT1/EXT2 and CHSY3/CHPF and 100 equivalents of each of the UDP-sugars required for chain elongation. At different time points, the products were separated by SDS-PAGE and detected by in-gel fluorescence. In the lower molecular weight range, SDS-PAGE resolved distinct bands that likely represent single sugar additions. EXTL3 was capable of adding a few sugars to the primed linker but failed to produce a polymer chain (Supplementary Fig. 9d). In contrast, EXT1/EXT2 displayed distributive (non-processive) chain polymerisation, with products growing progressively over time (Fig. 4f). In agreement with the experiments described above, Xyl phosphorylation had no effect on HS polymerisation (Supplementary Fig. 9e). The length of the HS backbone synthesised in vitro was limited by the availability of UDP-sugar donors. When GlcNAc-TetraP-BETA-5-FAM was incubated with EXT1/EXT2 and varying amounts of sugar donors, the maximum product distribution grew in line with the donor amount (Supplementary

Fig. 9f). Our finding that EXT1/EXT2 is the HS polymerase agrees with previous results obtained in vitro[33–35] and in cells[11,19,36].

CHSY3/CHPF was a notably poorer polymerase than EXT1/EXT2, with the majority of products containing less than 10 sugars in addition to the linker tetrasaccharide (Fig. 4g). We note that the CHSY3/CHPF complex has been shown to produce much shorter chains than CHSY1-containing complexes[17].

## GAG modification of folded core proteins in vitro

Thus far, the acceptor substrates used in this study have been peptides. To establish whether folded proteins could also be modified using OPME synthesis, we expressed a soluble form of the glypican-1 core protein (sGPC1) in a xylosyltransferase-deficient CHO cell line, CHO pgsA-745. These cells indeed produced an unmodified core protein, and GAG biosynthesis could be restored by cotransfection with XT2 (Supplementary Fig. 10a), as shown previously for the biglycan core protein[37]. Unmodified sGPC1 was purified from the CHO pgsA-745 cell culture medium (Supplementary Fig. 10b) and incubated for 1 h with the GAG biosynthetic enzymes, UDP-sugars and ATP. Glycosylation of sGPC1 was analysed by SDS-PAGE and Western blotting against the C-terminal FLAG tag (Fig. 5b). Following incubation with the linker enzymes and their respective substrates, sGPC1 ran at a higher molecular weight, indicating that linker tetrasaccharides had been added to some or all of the three Ser-Gly sequons in the core protein. Additional incubation with EXTL3 or CSGALNACT2 led to a barely detectable further increase in molecular weight. Finally, incubation with all enzymes, including the polymerases, led to a high molecular weight smear with EXT1/EXT2 but not with CHSY3/CHPF, mirroring the specific modification of glypican-1 with HS chains in vivo (Fig. 5a). When the reactions were incubated overnight, the EXT1/EXT2 product increased further in size and there now was a detectable shift in the CHSY3/CHPF product as well, indicating polymerisation of a CS backbone (Supplementary Fig. 10c).

An analogous experiment was done with decorin, a secreted PG that is modified with a single CS/DS chain in vivo. Incubation of decorin core protein produced in xylosyltransferase-deficient CHO cells with

the linker enzymes, priming enzymes and polymerases resulted in a high molecular weight smear with CSGALNACT2 and CHSY3/CHPF but not with EXTL3 and EXT1/EXT2. Although polymerisation of the CS backbone was already apparent after 1 h (Fig. 5b), a pronounced smear only emerged after overnight incubation (Supplementary Fig. 10d). No HS backbone was detected even after overnight incubation, suggesting that EXTL3 had failed to prime the linker tetrasaccharide on decorin.

## Discussion

OPME synthesis of near-authentic glycopeptide substrates has allowed us to study the initiation steps of HS and CS biosynthesis, which previously were inaccessible to enzymological analysis. Our results suggest a straightforward mechanism of GAG specification: (i) the enzymes initiating CS synthesis (CSGALNACTs) recognise the linker tetrasaccharide but not the polypeptide chain; they, therefore, are promiscuous with regard to the core protein; (ii) the enzyme initiating HS synthesis (EXTL3) makes only a few interactions with the linker but recognises specific features of the polypeptide chain; it, therefore, is specific for certain core proteins. In other words, CS is the default modification, and HS needs to be specified. This mechanism was predicted nearly 30 years ago, based solely on an analysis of the GAG chain type added to various sequences in cells[23].

We found that EXTL3 specificity is determined by a basic exosite >20 Å from the GlcNAc transferase site. Enzyme-substrate interactions at this exosite appear to be quite specific, as shown by the wide range of catalytic efficiencies ($k_{cat}/K_M$ values) for bona fide EXTL3 substrates. For example, the syndecan-2 and betaglycan glycopeptides have similar apparent $K_M$ values, but their $k_{cat}$ values differ by three orders of magnitude (Fig. 2b). We think that the linker tetrasaccharide of the syndecan-2 peptide may not be presented optimally to the GlcNAc transferase site, resulting in low catalytic efficiency. In contrast, CSGALNACT2 had very similar catalytic efficiency for all substrates tested, which is expected if only the tetrasaccharide portion of glycopeptides is recognised by the enzyme.

Further evidence for specific EXTL3-substrate interactions at the basic exosite comes from the analysis of single-residue substitutions in the acceptor polypeptide. In betaglycan, the substitution of the aspartic acid in position +7 abolished HS modification in CHO cells[21] and dramatically reduced priming by EXTL3 in our experiments (Fig. 3a). The milder effect of substituting the tryptophan in position +2 is more difficult to explain, given that the EXTL3 exosite does not contain any obvious hydrophobic patches. We hypothesise that tryptophan may form a stacking interaction with one of the arginines of EXTL3. Such an interaction could also explain the increased synthesis of HS on xylosides with hydrophobic aglycones[38]. GAG attachment sites that are not modified with HS (e.g. CSPG4 and decorin) also contain acidic residues, but the number, position or spacing of these residues must be incompatible with productive binding to the EXTL3 exosite. Structures of EXTL3-glycopeptide complexes will be required to fully define the signature of attachment sites modified with HS in vivo.

In the living cell, the biosynthetic outcome (HS or CS) may be determined by spatial segregation of the enzymes or their competition within the same compartment. In the former scenario, core proteins would have to encounter EXTL3 before CSGALNACTs in order to prevent all PGs from becoming modified with CS. If the two initiating enzymes were competing for substrates, the low $K_M$ values of bona fide EXTL3 substrates are predicted to result in preferential initiation of HS against a constant background CSGALNACT activity. The exact HS/CS ratio at a given Ser-Gly sequon would then be influenced by the relative amounts of EXTL3 and CSGALNACTs, the amounts of their respective UDP-sugar donors, the amount and transit time of the core protein, co-localisation of biosynthetic enzymes and membrane-bound core proteins, and so on. We believe that any HS site can be modified with CS

(for example, the N-terminal region of perlecan is modified solely with HS or with an HS/CS mixture depending on cell type[39–41]). However, we predict that CS sites may rarely, if ever, be modified with HS, given the very low activity of EXTL3 for Ser-Gly sequons lacking an HS signature.

Genetic deletion of FAM20B in cells severely reduces the total amount of GAG chains, and *FAM20B* mutations in humans cause lethal neonatal short limb dysplasia with multiple dislocations. In agreement with earlier studies[13,42,43], we found that FAM20B-mediated phosphorylation of the linker tetrasaccharide enhances the reactions catalysed by B3GALT6, B3GAT3 and CSGALNACT1. Additionally, we found that phosphorylation enhances the reactions catalysed by EXTL3 and CSGALNACT2. Thus, phosphorylation affects all steps in linker maturation after the Gal-Xyl-protein intermediate. However, we observed no effect of phosphorylation (positive or negative) on the elongation of EXTL3-primed linkers by EXT1/EXT2, contrary to a study that suggested that phosphorylation is an inhibitory modification that needs to be removed before HS elongation can occur[44]. Dephosphorylation of the linker is carried out by the phosphatase XYLP[45]. Genetic deletion XYLP had no effect on GAG biosynthesis in cells, supporting our conclusion that phosphorylation does not inhibit elongation[11].

We found that HS polymerisation by soluble EXT1/EXT2 in vitro is not processive, in agreement with two recent studies[15,16]. We speculate that membrane-bound polymerases (as they exist in the Golgi compartment) may become processive above a certain density in the membrane, which could facilitate the efficient handover of the growing HS chain from one polymerase to another. Such a mechanism would be particularly efficient for the many proteoglycans that are membrane proteins themselves, e.g. syndecans and glypicans.

An element currently missing from our reconstituted system is the NDST enzymes, which create the *N*-sulfated domains in HS and are believed to act in concert with EXT1/EXT2[46]. Given that we observe the rapid emergence of long chains in the absence of any chain-modifying enzymes, NDSTs are not required for the elongation of the HS backbone. Supporting this conclusion, the combined genetic deletion of NDST1 and NDST2 in cells had a major effect on the sulfation level of HS, as expected, but did not alter the length of HS chains[19].

In overnight experiments, EXTL3 displayed GlcA transferase activity comparable to EXT1/EXT2 for addition of the first GlcA to the primed linker tetrasaccharide (Fig. 4b), but only EXT1/EXT2 was able to polymerise a long HS backbone (Supplementary Fig. 9d). The GlcA transferase activity of EXTL3 is surprising as the active site of the N-terminal GT47 domain is predicted to be unable to bind UDP-GlcA[28]. We do not believe that the EXTL3 GlcA transferase activity seen in our experiments is due to contamination by EXT1/EXT2[28], given the purity of our protein preparations (Supplementary Fig. 1a) and the comparable amounts of product obtained with EXTL3 and EXT1/EXT2. We also think that it is unlikely that a secreted EXTL3 protein would form a stable complex with the transmembrane forms of EXT1 or EXT2 that are present in the cells used for protein production. Additional experiments would be needed to pinpoint the origin of EXTL3's GlcA transferase activity, but because this question has no bearing on any of our conclusions, we did not investigate the matter further. The failure of EXTL3 to polymerise a long HS backbone is likely to be the result of weak acceptor binding at the GlcNAc transferase site: without assistance from the exosite, longer acceptor glycans are likely to have very high $K_M$ values, preventing efficient elongation.

In summary, we have developed a powerful in vitro system that has allowed a rigorous enzymological examination of GAG chain selection. The current system covers all steps up to and including chain elongation. In the future, chain-modifying enzymes may be added, perhaps allowing the in vitro assembly of biologically active GAG chains. It should also be possible to incorporate chemically modified sugars, either using native or engineered glycosyltransferases and thereby generate novel glycoconjugates for discovery research.

## Methods

### Cloning and mutagenesis

Unless otherwise stated, all enzymes used for DNA cloning were from New England Biolabs. DNA encoding the lumenal domains of GAG biosynthetic enzymes (see Supplementary Table 1 for UniProt IDs and construct boundaries) was amplified from human cDNA clones (Horizon Discovery) using Q5 DNA polymerase and ligated using T4 DNA ligase (Thermo Fisher) into modified pCEP-Pu vectors[47]. 5' of the insert of interest, vector-derived sequences encode the BM-40 signal peptide, a FLAG-tag (EXT1) or His$_6$-tag (all other constructs), a tobacco etch virus protease cleavage site and an optional maltose binding protein (B3GALT6 and CSGALNACT1 only)[48]. DNA encoding residues 1–529 of human glypican-1, including the native Kozak sequence, was amplified from a cDNA clone (Horizon Discovery) and cloned into the pcDNA3.1(+) vector (Invitrogen); a sequence coding for a C-terminal FLAG tag was introduced by the reverse PCR primer. DNA encoding full-length human XT2 was cloned into the pcDNA3.1(+) vector in the same way but without adding a tag. The expression plasmid for human FLAG-tagged decorin was purchased from GenScript (OHu16408D). Subcloning efficiency DH5α competent cells (Thermo Fisher) were transformed by heat shock and used for plasmid amplification. Qiagen Plasmid Plus Miniprep, Maxiprep or Gigaprep kits were used for plasmid purification. All constructs were verified by DNA sequencing (Genewiz).

Mutagenesis of EXTL3 was carried out using overlap extension PCR[49]. The EXTL3-K914A/K917A and EXTL3-K905A/R907A constructs were made first. The double point mutations were introduced together. The EXTL3-K914A/K917A construct was used as a template to produce EXTL3-K905A/R907A/K912A/K914A/K197A. The three additional point mutations were introduced together. The final PCR products contained *Nhe*I and *Xho*I restriction sites for ligation into the modified pCEP-Pu vector.

### Enzyme expression and purification

Expi293F cells (Thermo Fisher, A14635) were cultured in suspension in Corning Erlenmeyer cell culture flasks in FreeStyle™ expression medium (Thermo Fisher) at 37 °C, 8% $CO_2$ and 125 rpm in a New Brunswick S41i CO2 incubator (Eppendorf). For transfection, cells were grown to a density of $10^6$ cells/mL. The cells were centrifuged at 100 g for 4 min, resuspended in fresh cell culture medium, and transfected with plasmid DNA using polyethylenimine (PEI) MAX 40 kDa (Polysciences). The DNA:PEI ratio was 1:3, and 1 µg of DNA was used per $10^6$ cells. For co-transfection of plasmids coding for obligate heterodimers (EXT1/EXT2 and CHSY3/CHPF), 500 ng of each plasmid was used. DNA and PEI were diluted separately and incubated for 5 min in Opti-MEM™ with GlutaMAX (Thermo Fisher) in 1/20 of the cell culture volume to be transfected. The PEI and DNA mixtures were then combined and incubated at room temperature for 30 min before being added to the cells. The transfected cells were incubated for 5 days at 37 °C, 8% $CO_2$ and 125 rpm.

The cell culture medium was centrifuged at 4000g for 20 min and then filtered through a 0.45 µm cellulose acetate filter (Thermo Fisher). The pH of the filtered medium was adjusted by adding 1 M Na-HEPES pH 7.5 (Sigma Aldrich) to a final concentration of 25 mM. For all protein constructs except XT1 and the FLAG-EXT1/His-EXT2 complex, a two-step purification was carried out. The cleared medium was loaded onto a 1 or 5 mL HisTrap Excel column (Cytiva) at 1 mL/min and 4 °C, using an Äkta pure chromatography system (Cytiva). The column was washed with 20 column volumes of 25 mM Na-HEPES pH 7.5, 150 mM NaCl, 20 mM imidazole (Sigma Aldrich), and bound protein was eluted with 10 column volumes of 25 mM Na-HEPES pH 7.5, 150 mM NaCl, 500 mM imidazole. The eluate was concentrated to 500 µL using a 10 kDa molecular-weight cut-off Vivaspin filtration device (Sartorius), and further purified on a Superdex 200 Increase 10/300 size exclusion column (Cytiva), using 25 mM Na-HEPES pH 7.5, 150 mM NaCl as the running buffer. Fractions containing protein were concentrated to 2-10 mg/mL and snap-frozen in liquid nitrogen. The final protein yields ranged from 1 to 10 mg per L of cell culture medium. Protein concentrations were determined by measuring $A_{280}$ using a NanoDrop spectrophotometer (Thermo Fisher), and extinction coefficients calculated from the protein sequence (https://web.expasy.org/protparam).

The FLAG-EXT1/His-EXT2 complex was produced by co-transfection of Expi293F cells with pCEP-FLAG-EXT1 and pCEP-His-EXT2 plasmids at an equal ratio. The medium was cleared as above and incubated with Pierce anti-DYKDDDK affinity resin (Thermo Fisher) for 1 h with rotation. The beads were washed with 30 bed volumes of 25 mM Na-HEPES pH 7.5, and 150 mM NaCl, and bound protein was eluted with 5 bed volumes of 100 µg/mL FLAG peptide (Sigma Aldrich) in washing buffer. The eluted protein was concentrated to 500 µL and further purified by size exclusion chromatography, as above. XT1 was purified using a three-step procedure as described previously[12].

### Core protein expression and purification

Soluble glypican-1 (sGPC1) and full-length decorin, both with a C-terminal FLAG tag, were expressed in a xylosyltransferase-deficient pgsA-745 Chinese hamster ovary cell line (ATCC, CRL-2242). Cells were grown at 37 °C, 5% $CO_2$ to 100% confluency in 175 cm$^2$ Nunc EasYFlasks™ in Dulbecco's modified Eagle's medium (DMEM)/F-12 (Invitrogen) supplemented with 1% (v/v) GlutaMAX (Thermo Fisher) and 10% (v/v) foetal bovine serum (FBS; Thermo Fisher). Cells were detached using 0.25% (v/v) Tyrpsin-EDTA (Thermo Fisher), diluted in fresh full medium, and distributed between 5 × 175 cm$^2$ flasks (40 mL/flask). The cells were then transfected in suspension with 4 mL of DNA/PEI (40 µg DNA:120 µg PEI) diluted in OptiMEM (as described above) per flask. The cells were allowed to attach for 5–6 h before the medium was replaced with DMEM-F12, 1% (v/v) GlutaMAX, and 5% (v/v) FBS. The transfected cells were incubated for 5 days at 37 °C, 5% $CO_2$. The cleared cell culture medium was incubated with anti-DYKDDDK beads, and the protein was eluted using a FLAG peptide as described above. FLAG peptide was removed by repeated rounds of concentration and dilution in 25 mM Na-HEPES pH 7.5, and 150 mM NaCl using a 10 kDa molecular-weight cut-off Vivaspin filtration device. To restore xylosyltransferase activity in these cell lines, the XT2 plasmid was co-transfected with the core protein plasmid at an equal ratio.

### Peptide synthesis

All unmodified peptides were purchased from Genscript. The BETA-5-FAM peptide (5-FAM-ε-Ahx-SPGDSSGWPDGYEDLE) was synthesised by the peptide facility of the Francis Crick Institute as follows. Solid phase synthesis of the peptide took place on an automated peptide synthesiser (Activotec P11) using Rink Amide AM resin (0.1 mmol; Merck) and N(α)-Fmoc amino acids, including Fmoc-εAHx-OH as appropriate. HATU was used as the coupling reagent with a fivefold excess of amino acids. After completion of chain assembly, the peptide was dye-labelled using a solution of 5-FAM (4 equivalents) in 1:1 dimethyl-sulfoxide:N-methyl-2-pyrrolidone (NMP). N,N-diethylpropylethylamine (4 eq) was added, followed by Oxyma Pure (4 eq) in NMP. After 3 min, N,N'-diisopropylcarbodiimide (4 eq) was added, and then after 30 min, the solution was added to the resin and allowed to react overnight at room temperature. The resin was next washed with dimethylformamide (DMF), followed by dichloromethane (DCM), treated with piperidine and washed again with DMF and DCM. The peptide was cleaved from the resin, and protecting groups were removed by the addition of a cleavage solution (95% trifluoroacetic acid (TFA), 2.5% $H_2O$, 2.5% triisopropylsilane). After 2 h, the resin was removed by filtration, and the peptide precipitated with diethyl ether on ice. The peptide was isolated by centrifugation, then dissolved in water and freeze-dried overnight. Portions of the peptide were purified on a C8 reverse phase HPLC column (Agilent PrepHT Zorbax 300SB-C8,

21.2×250 mm, 7 m) using a linear solvent gradient of 0–30% acetonitrile with 0.08% TFA) in H$_2$O with 0.08% TFA over 40 min at a flow rate of 8 mL/min. The purified peptide was analysed by liquid chromatography–MS (LC–MS) on an Agilent 1100 LC–MSD instrument.

## One-pot multienzyme synthesis of glycopeptides

Glycans were assembled on the peptides using one-pot multienzyme (OPME) reactions containing all the requisite enzymes, UDP-sugars and ATP. Reactions were carried out in 25 mM MnCl$_2$, 50 mM Na-HEPES pH 7.5, and 50 mM NaCl, in a total reaction volume of 20–30 μL (analytical scale) or 0.5–2 mL (preparative scale). Enzymes were added at 0.025 μg/mL, peptide at 500 μM and UDP-sugars at a two-fold molar excess to the acceptor. For example, the GlcA-Gal-Gal-Xyl2P-peptide (TetraP-peptide) was synthesised from 500 μM peptide, 1 mM UDP-Xyl (Carbosynth), 2 mM UDP-Gal (Carbosynth), 1 mM UDP-GlcA (Sigma Aldrich) and 5 mM ATP (Sigma Aldrich) using 0.025 μg/mL each of XT1, B4GALT7, MBP-B3GALT6, B3GAT3 and FAM20B. For the Tetra-peptide, FAM20B and ATP were omitted. For GlcNAc-Tetra(P) and GalNAc-Tetra(P) peptides, 0.025 μg/mL EXTL3 and 1 mM UDP-GlcNAc, or 0.025 μg/mL CSGALNACT2 and 1 mM UDP-GalNAc (Sigma Aldrich), were added additionally. The mixtures were left at 30 °C overnight, and reaction progress was monitored by HPLC using a Poroshell 120 EC-C18 column (Agilent) and an Agilent 1260 Infinity II system with Chemstation software (Agilent). Peptides were separated using a gradient of acetonitrile with 0.085% (v/v) TFA (Thermo Fisher) against Chromplete water (Thermo Fisher) with 0.1% (v/v) TFA. Typically, a gradient of 10–30% acetonitrile over 17 min or 15–30% acetonitrile over 25 min, at 0.5 mL/min, was used. Peptides were detected using UV absorbance at 280 or 214 nm. Peak fractions were collected and analysed by MS (see below).

Following HPLC confirmation, the glycopeptides were purified on a Superdex 30 Increase 10/300 size exclusion column (Cytiva) with 0.1 M ammonium acetate pH 4.75 as the running buffer. Glycopeptides were desalted using a Pierce C18 Spin Column (Thermo Fisher), eluted with 50% acetonitrile and lyophilised using a Savant SpeedVac SPD120 Vacuum Concentrator. Lyophilised peptides were resuspended in 50 mM Na-HEPES pH 7.5 and 50 mM NaCl for biochemical analysis.

## Glycosyltransferase assays

To study the initiation step, Tetra- and TetraP-peptides were incubated at 100 μM in 5 mM MnCl$_2$, 50 mM Na-HEPES pH 7.5, 50 mM NaCl, with 200 μM UDP-GlcNAc or UDP-GalNac, and 0.5 μM EXTL3, EXT1/EXT2, CSGALNACT2, or CHSY3/CHPF, overnight at 30 °C. The reaction products were analysed by HPLC, as above, followed by MS. Similar assay conditions were used for assessing the effect of Xyl phosphorylation on B3GALT6 activity. Gal-Xyl-BKN or Gal-Xyl2P-BKN at 100 μM was incubated with 0.2 μM MBP-B3GALT6 and 200 μM UDP-Gal, in an overnight reaction at 30 °C.

To determine kinetic parameters, the UDP-Glo™ assay (Promega) was used. Reactions were performed in 96-well white assay plates (Greiner) in a 25 μL reaction volume. All reagents were diluted in 5 mM MnCl$_2$, 50 mM Na-HEPES pH 7.5 and 50 mM NaCl. The indicated glycopeptide was serially diluted (2-fold), giving a 1.56–100 μM concentration range. A fixed concentration of the indicated glycosyltransferase (8–50 nM) and associated UDP-sugar (100 μM; Promega) were added to the plate to initiate the reactions at 0, 15 and 30 min, and the plates incubated at room temperature. The reactions were stopped immediately after the 30 min time point by the addition of the UDP-Glo™ nucleotide detection reagent and incubated at room temperature for 1 h. Luminescence was measured using a Tecan Spark plate reader with Spark Control V2.3 software. Relative light units were converted to UDP concentration using a UDP standard curve, which was made according to the manufacturer's instructions. UDP production was linear over the time course in all reactions. Initial rates were

calculated through linear regression and fitted with the Michaelis-Menten equation using GraphPad Prism 9.

## HS and CS polymerisation assays

The indicated glycans were assembled on the BETA-5-FAM peptide and the resulting glycopeptides purified, as described above. The glycopeptides were incubated at 100 μM in 50 mM Na-HEPES pH 7.5, 50 mM NaCl and 25 mM MnCl$_2$, with 0.5 μM of the indicated glycosyltransferase (EXTL3, EXT1/EXT2, or CHSY3/CHPF), 10 mM UDP-GlcA, and 10 mM UDP-GlcNAc or UDP-GalNAc, for the indicated time, in a total reaction volume of 10–30 μL. The reactions were stopped by the addition of sodium dodecyl sulfate (SDS)-sample buffer and the samples were heated at 90 °C for 20 s. The reaction products were separated using a 20% SDS-PAGE (polyacrylamide gel electrophoresis) gel. In-gel fluorescence was recorded using a Fujifilm FLA-5000 image analyser (Raytek) with excitation at 473 nm and detection at 530 nm. Images were produced in ImageJ using the ISAC Manager plug-in.

sGPC1 or decorin purified from pgsA-745 Chinese hamster ovary cells was incubated at 2 μM in 25 mM MnCl$_2$, 50 mM Na-HEPES pH 7.5, 50 mM NaCl, with 0.015 μg/mL of the linker-synthesising enzymes (XT1, B4GALT7, B3GALT6, B3GAT3, FAM20B), 0.5 mM UDP-Xyl, 1 mM UDP-Gal, 1 mM UDP-GlcA and 3 mM ATP, in a total reaction volume of 10 μL. For HS or CS initiation, 0.5 μM EXTL3 and 0.5 mM UDP-GlcNAc, or 0.5 μM CSGALNACT2 and 0.5 mM UDP-GalNAc, were added along with the linker enzymes, UDP-sugars and ATP. For HS and CS polymerisation, 0.5 μM His-EXT1/EXT2 or CHSY3/CHPF was additionally added to the reactions. The samples were incubated for 1 h at 30 °C before boiling in an SDS-sample buffer for 3 min. Samples were separated on a 10% SDS-PAGE gel and then transferred to nitrocellulose (Bio-Rad) using a Trans-Blot Turbo (Bio-Rad). The membrane was blocked for 1 h in Intercept Blocking Buffer (LI-COR Biosciences) before being incubated overnight at 4 °C with mouse monoclonal anti-FLAG M2 antibody (Sigma Aldrich, F3165, diluted 1:200 in Intercept Blocking Buffer with 0.2% (v/v) Tween). The membranes were washed 4 times for 5 min in phosphate-buffered saline with 0.1% (v/v) Tween (PBS-T; Sigma Aldrich) and incubated with IRDye800CW donkey anti-mouse secondary antibody (LI-COR, 926–32212, diluted 1:15000 in Intercept Blocking Buffer with 0.2% (v/v) Tween) for 1 h at room temperature. Membranes were washed a further 4 times for 5 min in PBS-T before being imaged using an Odyssey FC imaging system in the 800 nm channel (LI-COR). Image generation was done using Image Studio Lite (LI-COR; V5.2).

## Mass spectrometry

MALDI-TOF (matrix-assisted laser desorption/ionisation-time of flight) MS was carried out on all glycopeptides used in this study. HPLC peak fractions were diluted 1:1 with 3,4-diaminobenzophenone dissolved at 10 mg/mL in 75% (v/v) acetonitrile or with 2,5-dihydroxybenzoic acid dissolved at 20 mg/mL in 70% (v/v) methanol. The glycopeptide/matrix mix (1 μL) was then spotted on a 384-well target plate (AB Sciex) and left to dry at room temperature overnight. The following day samples were analysed in positive or negative-ion mode using a 4800 Plus MALDI–TOF/TOF Analyser (AB Sciex) with 4000 Series Explorer (Applied Biosystems). Tandem MS of glycopeptides was carried out using collision-induced dissociation fragmentation. Data were extracted with the Data Explorer software (Applied Biosystems), and semi-manual data analysis was carried out with the open-source software Skyline (MacCoss Lab). For final graphic editing, CorelDraw software was used.

## Size exclusion chromatography with multi-angle light scattering

CSGALNACT2 ΔCC at a concentration of 4 mg/mL was injected onto a Superdex 200 Increase 10/300 column (Cytiva) connected to an Agilent 1260 Infinity system. The running buffer was 25 mM Na-HEPES pH 7.5, 150 mM NaCl and the flow rate was 0.5 mL/min. Light scattering

and refractive index changes were monitored using in-line Wyatt Mini Dawn and Optilab T-rEX detectors (Wyatt Technology Corp). The data were analysed with the Wyatt ASTRA V software and gave an experimental mass of 109 kDa.

## Crystal structure determination

For crystal screening, a truncated EXTL3 construct lacking the N-terminal coiled-coil region was produced (EXTL3 ΔCC, residues 154–919). The protein was expressed and purified as described above, except that 20 mM bicine pH 8.6, 150 mM NaCl was used as the running buffer in size exclusion chromatography[50]. The protein was concentrated at 6.8 mg/mL. Crystal screens were set up using a Mosquito robot (SPT Labtech) and a range of commercial screens in 96-well MRC sitting drop plates (Molecular Dimensions and Hampton Research) by mixing 200 nL of protein with 200 nL of reservoir solution. The plates were incubated at 16 °C. Cuboid crystals were obtained in several conditions, which typically contained 12.5–20% (w/v) PEG3350 and 0.1–0.2 M of dicarboxylic acid. The apo-enzyme structure was solved using crystals grown in 12.5% (w/v) PEG3350 (Sigma Aldrich), 0.1 M sodium malonate (Sigma Aldrich), pH 7.0. The crystals were cryoprotected with 30% (v/v) glycerol and frozen in liquid nitrogen. X-ray diffraction data were collected at 100 K on the microfocus beamline I24 at the Diamond Light Source. A high-redundancy dataset was collected from a single crystal and processed using the automatic XIA2-DIALS pipeline[51,52]. The resolution limit was determined using the $CC_{1/2}$ criterion[53] as implemented in AIMLESS[54]. The structure was solved by molecular replacement using PHASER[55] and Protein Data Bank (PDB) entry 7AU2[28] as a search model. Model building and refinement were done using COOT[56] and PHENIX[57]. Crystals grown in 12.5% PEG3350, 0.1 M malic acid (Sigma Aldrich) pH 7.0 were soaked in 15% PEG 3350, 0.1 M malic acid pH 7.0, 4.5 mM TetraP-BETA glycopeptide, 5 mM UDP (Sigma Aldrich) and 5 mM $MnCl_2$ (Sigma Aldrich) overnight at 16 °C. The soaked crystals were cryoprotected and frozen, as described above. Diffraction data were collected on beamline I04-1 at Diamond Light Source and processed as described above. The UDP and $Mn^{2+}$ ions were readily apparent in difference maps, but no density was observed for the glycopeptide.

## Modelling of enzyme-substrate complexes

Model building was done manually in COOT. The AlphaFold model of B3GALT6 (Q96L58) was superimposed with B3GNT2 structures to obtain the positions of the UDP-sugar donor (PDB:7JHL[58]) and a disaccharide acceptor (PDB:7JHM[58]). The substrates were changed to UDP-Gal and Galβ1-4Xyl, respectively. No changes were made to the protein structure.

To model the glycopeptide acceptor into the EXTL3 crystal structure, the structure of EXTL2 containing UDP and a GlcAβ1-3Galβ1-O-naphthalenemethanol acceptor (PDB: 1ON8[29]) was superimposed onto the EXTL3. The naphthalenemethanol was replaced with Gal using the Galβ1-3Gal substrate of a B3GAT3 structure (PDB:3CU0[42]) to produce GlcAβ1-4Galβ1-3Galβ. Xyl was added using Galβ1-4Xyl from the B3GALT6 model described above to produce the complete linker tetrasaccharide. The torsion angles of the glycosidic bonds were adjusted to their mean values in an NMR structure of the linker tetrasaccharide[30]. A phosphate group was added to Xyl, and torsion angles were adjusted to minimise steric clashes. Finally, a serine was added to Xyl using the glycopeptide acceptor from a xyloside-α1,3-xylosyltransferase structure (PDB:4WM0[59]). The EXTL3 AlphaFold model (O43909) was used to fill in a disordered active site loop in the crystal structure (residues 864–868). No other changes were made to the protein structure.

ColabFold[31] was used to obtain the structure of a CSGALNACT2 homodimer, which was superimposed with a B4GALT7 structure (PDB:4M4K[60]) to obtain the positions of the UDP-sugar donor and a disaccharide acceptor, xylobiose. The phosphorylated tetrasaccharide

from the EXTL3 model described above was docked into the CSGAL-NACT2 active site by superimposing the terminal GlcA with the non-reducing Xyl of PDB:4M4K. A clash of the GlcA carboxylate with Arg428 was relieved by changing the torsion angles of the Arg428 side chain. No other changes were made to the protein structure.

## Reporting summary

Further information on research design is available in the Nature Portfolio Reporting Summary linked to this article.

## Data availability

All data described in the manuscript and the Supplementary Information are available. The crystallographic data generated in this study have been deposited in the Protein Data Bank under accession codes 8OG1 (EXTL3 apo structure) and 8OG4 (EXTL3 UDP complex). The annotated mass spectra relating to Supplementary Table 2 are provided as Supplementary Data. Source data are provided in this paper. Source data are provided in this paper.

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

## Acknowledgements

This research was funded by a Wellcome Senior Investigator Award to E.H. (101748/Z/13/Z) and a Project Grant from the UK Biotechnology and Biological Sciences Research Council to E.H. and B.S. (BB/T01279X/1). This work was supported by the Francis Crick Institute, which receives its core funding from Cancer Research UK (CC2127), the UK Medical Research Council (CC2127) and the Wellcome Trust (CC2127). We acknowledge the use of the crystallisation facility at Imperial College London, which is supported by the Biotechnology and Biological Sciences Research Council (BB/D524840/1) and Wellcome Trust (202926/Z/16/Z). We acknowledge Diamond Light Source for beamtime under proposal MX31800. We thank Saffi Hussain for help with protein production and the many students who helped with cloning and preliminary experiments, especially Himani Amin, Yichen Lu, Simona Gromovaite and Pranav Bhamidipati. We thank the peptide facility at the Francis Crick Institute for the synthesis of the 5-FAM-labelled peptide. For the purpose of Open Access, the authors have applied a CC BY public copyright licence to any Author Accepted Manuscript version arising from this submission.

## Author contributions

D.S., S.M.H., B.S., D.C.B. and E.H. planned the experiments; D.S., M.B.-W. and D.C.B. produced the recombinant enzymes; M.B.-W. devised the one-pot multienzyme strategy; D.S. performed the biochemical experiments; A.K. and D.S. performed the mass spectrometry experiments; R.M.M., D.S., E.H. performed the crystallographic experiments and molecular modelling; D.S. and E.H. wrote the paper with input from all authors.

## Competing interests

The authors declare no competing interests.
