## [Peer Review File · Nature Communications]

REVIEWER COMMENTS

Reviewer #1 (Remarks to the Author):

This paper is a long awaited in vitro study of enzymes involved in assembly of the linkage region and the backbone structures of heparan sulfate and chondroitin sulfate chains. The authors should be commended on a detailed, well thought-out and well-presented set of studies. Although several of the findings confirm earlier work performed in cell culture models, the application of one-pot multienzyme (OPME) reaction strategy described here led to several novel findings and opens up the system for more detailed structural and functional analysis of glycosaminoglycan assembly. I wish that I was a coauthor instead of a reviewer of this paper.

I have only a few minor comments for the authors to consider.

1. Add Type XVIII collagen to the list of secreted proteoglycans in the Introduction.
2. Line 74, substitute "heparan sulfate" for "GAG chains". (CS does not undergo N-deacetylation).
3. Paragraph starting on Line 108. If you have additional information on the structures of other HSPGs (e.g., Sdc1, perlecan) please add that to the supplement (Fig. S2) or add a comment that analysis of the other proteoglycans shows that the GAG chains are added in unstructured domains in other proteoglycans.
4. How do you explain the observation that TetraP-SDC2 is a good substrate for CSGALNACT2? You might comment here that the native proteoglycans might behave differently.
5. Fig. 5 shows that EXTL3 does not appear to polymerize chains on sGPC1, but does it on GPC1 peptide substrates? That would get at this question of native structures vs. peptide substrates.
6. What happens if you mix CSGALNACT2 and EXTL3 with SDC4 and GPC1 peptide substrates. Perhaps the promiscuity of the sites disappears.
7. Line 239-240. Please comment (here or somewhere) on xylosides, which were originally shown to prime CS by default, and a combination of HS and CS if the aglycone was more hydrophobic (PMID 8276811 and others). This observation led to the idea that the core proteins for HS might contain hydrophobic amino acid(s) in sequences flanking the attachment site that are preferentially recognized by EXTL3.
8. In the discussion, please comment that the in vivo two-dimensional arrangement and concentration of the enzymes might affect the processing of the chains.
9. Regarding the role of EXTL2, do the authors really think its involved in HS assembly? . The alpha-GalNAc transferase activity is 10-times that of alpha-GlcNAc transferase activity and this reviewer is not aware of any data demonstrating an alpha-GalNAc residue on a linkage pentasaccharide. The OPME system could be used to test this directly.

10. Can you comment in the Discussion on perlecan, which is typically an HSPG, but in chondrocytes, can contain CS (PMID:17169545 and PMID:20507176). Do the glycosylation sites conform to the arrangement described for GPC1 or to SDC2?

Reviewer #2 (Remarks to the Author):

In this paper the authors have interrogated the mechanism of how heparan sulfate (HS) and chondroitin sulfate (CS) glycosylation sites are distinguished on target proteins. Both glycosaminoglycans are typically attached to Ser-Gly sequons in unstructured regions of proteins, but the mechanism for the bias observed for HS or CS at certain sites has been less clear. The authors developed an enzymatic one-pot HS or CS synthesis system to test the specificity of both pathways using peptides with the amino acid sequences from target proteins that are known to carry either or both glycosaminoglycans in vivo. Using HPLC or gel-based fluorescence assays, the authors show CS is synthesized with little bias to the peptide sequence. In contrast, HS synthesis efficiency is improved with the presence of acidic residues C-terminal to the glycosylation sequon. These observations are rationalized using an X-ray crystal structure and AlphaFold predictions with substrates placed based on structural alignments with other published structures. The authors predict CSGALNACT2, which initiates CS synthesis, to mostly bind the acceptor sugar rather than the peptide, providing a possible cause for the promiscuity of CS addition with respect to the peptide sequence. In contrast, the X-ray crystal structure of EXTL3, which initiates HS synthesis, shows an electropositive patch complementing the predicted binding site of the peptide, providing a possible mechanism for the peptide specificity observed for EXTL3. Additionally, the authors show that the efficiency of the synthesis of the precursor tetrasaccharide linker, common to both glycosaminoglycans, is increased by the phosphorylation of xylose at the C2 hydroxyl of the growing acceptor substrate. However, this phosphorylation does not affect extension of the common acceptor substrate during HS or CS synthesis.

This paper provides insight into aspects of the kinetics and the mechanism of two glycosylation pathways that are essential for a wide variety of cellular processes. The main conclusions of the paper are in line with earlier proposals and predictions and are generally supported by the data. Improvements in the wording in certain areas of the text and in the interpretation of some of the data are required.

Results: Fig 1A: It would help the reader if the figure indicated which peptides are primarily associated with either or both types of glycosaminoglycan. A more detailed discussion of why CSPG4 is primarily modified by CS (it has acidic residues C-terminal to the GAG sequon) would be helpful. Also, the authors should explain what allows BETA to carry both HS and CS despite featuring a similar acidic residue sequence to SDC4, a proteoglycan carrying predominantly HS. These anomalies should be

addressed in the discussion in the context of the proposed model of acidic residues C-terminal to the Ser-Gly sequon being important for HS synthesis.

Results: line 129. The authors investigate the effect of Xyl phosphorylation on the catalytic efficiency of B3GALT6 and B3GAT3 but do not comment on whether Xyl phosphorylation affects the catalytic efficiency of B4GALT7, the enzyme transferring a Gal sugar to the Xyl in question. Please comment on how Xyl phosphorylation affects the catalytic efficiency of B4GALT7.

Results: line 131. Suggest to directly refer to changes in catalytic efficiency rather than “increased the reaction” in the sentence, “We found that phosphorylation dramatically increased the reaction catalysed...”.

Fig. S3B: Graph for B3GALT6: Need to put phosphorylated and unphosphorylated substrate curves on different scales to be able to evaluate these data. Also, for the unphosphorylated substrates, accurate KM values cannot be accurately calculated when the maximum substrate concentration tested is at or below the KM. Data with higher substrate concentrations for these samples are needed for accurate KM values.

Fig. S3C Please include a panel with the AlphaFold model of B3GALT6 coloured by pLDDT.

Results: line 136. Propose to change “could be rationalised by structural modelling...” to reflect that an AlphaFold model was used and the substrate binding pocket was inferred by homology with a structure of a related enzyme.

Fig 2B: Accurate Michaelis-Menten constants cannot be calculated when the highest substrate concentration used is not even the concentration needed for half Vmax. Either repeat assays with a more appropriate range of substrate concentrations or mention that the KM cannot be accurately determined for these reactions.

Fig. S5B: Graph for B3GALT6: Need to put phosphorylated and unphosphorylated substrate curves on different scales to be able to evaluate these data. Also, especially for TetraP-Beta substrate, accurate KM values cannot be accurately calculated when the maximum substrate concentration tested is at or below the KM. Data with higher substrate concentrations for these samples are needed for accurate KM values.

Table S3: Please report unit cell angles, X-ray wavelength used, as well as average B-factors for the protein, waters and ligands.

Fig. S7A, please include a panel with the CSGALNACT2 ColabFold model coloured by pLDDT.

Results: For the models discussed in paragraphs starting on line 211 and 229, the structural insight would be substantially improved by including actual docking experiments where molecular dynamics simulations are used rather than manually placing substrates based on structural alignments. This is especially relevant for the paragraph starting on line 229 where specific molecular interactions are discussed. Given how these models were created, we suggest avoiding the discussion of specific residue-substrate interactions.

Reviewer #3 (Remarks to the Author):

Molecular mechanism of decision-making in glycosaminoglycan biosynthesis

In this manuscript, the authors perform a series of elegant experiments to dissect the mechanisms underlying the decision for glycosaminoglycan (GAG) enzymes to form heparan sulfate (HS) or chondroitin sulfate (CS) chains. Using an impressive one-pot multienzyme synthesis, Sammon et al. demonstrate the ability of GAG biosynthesis to proceed in vitro, though the degree of polymerization achieved is unclear, on a variety of proteoglycan (PG) peptide sequences. Using molecular modeling, the authors determine that a basic exosite in exostosin-like glycosyltransferase 3 (EXTL3) is responsible for the selective addition of HS GAGs, overriding the default CS. Through kinetic studies, the ability of glycopeptides to function as substrates for EXTL3 and CSGALNACT2 were elucidated, including the role of xylose phosphorylation in promoting GAG biosynthesis.

This manuscript represents a significant progress for the PG community, elucidating little-known molecular determinants of GAG attachment. Further, this multienzyme strategy provides a potential platform for the in vitro biosynthesis of GAGs. The authors' claims are generally well supported through the use of HPLC and MS analyses. However, there are a few areas of the manuscript that could be further elaborated.

Major points

1. Throughout the manuscript, the authors make a point that GAG attachment occurs in unstructured or disordered regions. I am curious whether the structure of these peptides/proteins varies between core proteins, and whether this could factor into the ability to be an EXTL3 substrate. Similarly, does the addition of a GAG chain change the conformation of such a peptide to impact its ability to be acted upon by EXTL3 and thus add a further determinant to HS/CS fate.

2. In lines 177-179, the authors state that SDC4 and GPC1 glycopeptides containing multiple closely spaced Ser-Gly sequons were both good substrates for EXTL3. Further, as these multiple sequons are suggested to be determinants of HS preference, providing further evidence to confirm that all SG sites in both proteins are occupied would be welcomed. Further, I postulate whether the mutagenesis of one or more of these SG sites would impact the identity of attached GAG chains. The authors refer to other PG peptides as evidence of single vs multi-site EXTL3 substrates not enhancing EXTL3 activity, but the amino acid sequences are quite different.

3. The authors allude to this one pot strategy as being a potential method for the in vitro generation of GAGs. Thus, I am curious about the degree of polymerization/chain length achieved in their experiments. This could be partially achieved by performing disaccharide analysis on the final glycopeptide or SEC analysis.

4. I would like to see evidence to support the author's claim that tetrasaccharide linkers had been added to all three Ser-Gly sites in their soluble GPC1 protein. MS has been used throughout the manuscript to support the author's claims of tetrasaccharide linker modification, performing such analyses on GPC1 would provide much-needed support to the claim that all three sites are modified.

a. If the "default" GAG chain is CS, I wouldn't expect to see a stunted CS backbone as the authors state in line 303. I wonder if the migration of HS and CS-modified GPC1 is simply different by SDS-PAGE, as the band appears to migrate slightly further than HS-modified GPC1 (further, in Fig. 3F and 3G, CS modified BETA migrated further than HS). A possible control for this would be native GPC1 enzymatically digested with heparinase and/or chondroitinase.

b. The GPC1 protein sequence possesses 6 Ser-Gly sites, I'm curious if the authors could postulate on why only three of these sites are typically modified?

5. The underlying discovery of the manuscript is the selective ability of EXTL3 to modify PG core proteins with HS chains instead of CS. As such, it would be nice to see the experiments performed in Figure 5 with a hybrid PG (such as Syndecan-1 or -3).

Minor points

1. In lines 50-52 it is unclear what function the authors are specifically referring to, presumably the binding of growth factors?

“... in the nervous system, for instance, HS is an attractive signal, whereas CS is a repellent”

2. The panels for Figure 4 are not referenced in order (A, D, E, B, C).

We thank all three reviewers for their detailed and constructive criticism. The additional experiments and the rewriting have improved our manuscript very substantially.

Reviewer #1

1. Add Type XVIII collagen to the list of secreted proteoglycans in the Introduction.

This has been done.

2. Line 74, substitute “heparan sulfate” for “GAG chains”. (CS does not undergo N-deacetylation).

This has been corrected.

3. Paragraph starting on Line 108. If you have additional information on the structures of other HSPGs (e.g., Sdc1, perlecan) please add that to the supplement (Fig. S2) or add a comment that analysis of the other proteoglycans shows that the GAG chains are added in unstructured domains in other proteoglycans.

We have added syndecan-1, perlecan, agrin, collagen XVIII and decorin to Supplementary Figure 2. Their GAG attachment sites are also unstructured. NetSurfP 3.0 is straightforward to use, so other sequences can be analysed by anyone who is interested. We have added a sentence on why we think that GAG attachment sites must be located in unstructured regions (pages 4-5 of the revised manuscript).

We are not aware of any experimental structure of a GAG attachment site in the Protein Data Bank. AlphaFold predictions are available for almost all proteoglycans and support our conclusion that GAG attachment sites are unstructured.

4. How do you explain the observation that TetraP-SDC2 is a good substrate for CSGALNACT2? You might comment here that the native proteoglycans might behave differently.

All TetraP-peptides are good substrates for CSGALNACT2. We think this is because the CSGALNACT2 interacts with the linker tetrasaccharide more extensively than EXTL3, and therefore does not rely on interactions with the acceptor polypeptide. The k_{cat}/K_M values of CSGALNACT2 for different acceptors are all quite similar (Figure 2b). The only outlier is TetraP-SDC4, which is a 5 to 10-fold better substrate than the other acceptors; this may be due to a fortuitous interaction with the substrate polypeptide.

5. Fig. 5 shows that EXTL3 does not appear to polymerize chains on sGPC1, but does it on GPC1 peptide substrates? That would get at this question of native structures vs. peptide substrates.

We did not test elongation by EXTL3 on the TetraP-GPC1 peptide, but we did so on the TetraP-BETA peptide: EXTL3 only added <10 sugars in an overnight reaction, even though

200 UDP-sugar equivalents were available (Supplementary Figure 9d of the revised manuscript). In contrast, EXT1/EXT2 polymerised a chain whose length appears to be limited only by the availability of UDP-sugars (new result in Supplementary Figure 9f). We have not done a time course of the EXTL3 reaction, but we expect only a few sugar additions within one hour, which may not be enough to result in a detectable shift of the ≈ 60 kDa sGPC1 glycoprotein.

6. What happens if you mix CSGALNACT2 and EXTL3 with SDC4 and GPC1 peptide substrates. Perhaps the promiscuity of the sites disappears.

We agree that a competition experiment would be interesting. However, the outcome *in vitro* could be tuned at will by altering the enzyme and substrate concentrations: EXTL3 would prevail at low acceptor concentrations because of its lower K_M values, and CSGALNACT2 would prevail at high acceptor concentrations because of its generally higher k_{cat} values. It would be difficult to relate these outcomes to the *in vivo* situation, where concentrations are unknown and other factors may play a role as well (see point 8). There is also the experimental complication that the products of EXTL3 and CSGALNACT2 (GlcNAc-TetraP-peptide and GalNAc-TetraP-peptide, respectively) cannot be distinguished by HPLC or mass spectrometry, so a new method for quantifying products would have to be devised.

In response to this and other comments, we have added a new paragraph to the Discussion on the relative priming of HS and CS in cells (page 13).

7. Line 239-240. Please comment (here or somewhere) on xylosides, which were originally shown to prime CS by default, and a combination of HS and CS if the aglycone was more hydrophobic (PMID 8276811 and others). This observation led to the idea that the core proteins for HS might contain hydrophobic amino acid(s) in sequences flanking the attachment site that are preferentially recognized by EXTL3.

The preferential priming of hydrophobic xylosides is now mentioned on page 12. To address this question experimentally, we have analysed a betaglycan peptide with the same Trp-to-Tyr substitution as used in PMID 8034692. This substitution indeed reduces priming by EXTL3, but to a much lesser extent than the Asp-to-Asn substitution or truncation of the peptide (revised Figure 3a). A speculative structural explanation of the effect has been added on page 12.

8. In the discussion, please comment that the *in vivo* two-dimensional arrangement and concentration of the enzymes might affect the processing of the chains.

Membrane localisation of the biosynthetic enzymes is now mentioned twice in the Discussion, once as a potential factor of HS/CS selection, and once in the context of processivity. This is an important question, which we intend to tackle in future studies, but it will require the development of new experimental approaches.

9. Regarding the role of EXTL2, do the authors really think its involved in HS assembly? The alpha-GalNAc transferase activity is 10-times that of alpha-GlcNAc transferase activity and

this reviewer is not aware of any data demonstrating an alpha-GalNAc residue on a linkage pentasaccharide. The OPME system could be used to test this directly.

We have tested the addition of GlcNAc and GalNAc to linker tetrasaccharides by EXTL2. In agreement with an earlier study (PMID 12562774), we observed very low specific activities (≈ 1000 -fold lower than EXTL3). Upon reflection, we feel that the discussion of EXTL2 adds little to our manuscript and have therefore deleted the paragraph on EXTL2.

10. Can you comment in the Discussion on perlecan, which is typically an HSPG, but in chondrocytes, can contain CS (PMID:17169545 and PMID:20507176). Do the glycosylation sites conform to the arrangement described for GPC1 or to SDC2?

The variable modification of perlecan is now mentioned as an example in the Discussion (page 13). We note that CS has also been detected on at least one other proteoglycan that is typically considered a HSPG, syndecan-4 (PMID 7520439). The N-terminal sites of syndecan-1 are also not exclusively modified with HS (PMID 8163535, PMID 7592967).

Reviewer #2

Results: Fig 1A: It would help the reader if the figure indicated which peptides are primarily associated with either or both types of glycosaminoglycan. A more detailed discussion of why CSPG4 is primarily modified by CS (it has acidic residues C-terminal to the GAG sequeon) would be helpful. Also, the authors should explain what allows BETA to carry both HS and CS despite featuring a similar acidic residue sequence to SDC4, a proteoglycan carrying predominantly HS. These anomalies should be addressed in the discussion in the context of the proposed model of acidic residues C-terminal to the Ser-Gly sequon being important for HS synthesis.

The HS/CS preferences of the GAG attachment sites studied have been added to Figure 1a.

In response to this point, and comments made by the other reviewers, we have expanded our discussion of EXTL3 specificity (pages 12 and 13 of the revised manuscript) and now acknowledge that a clear "HS signature" cannot currently be discerned.

It is also worth noting that all proteoglycans typically considered to be HSPGs may actually contain variable amounts of CS (see our response to point 10 of reviewer 1). This makes betaglycan less of a special case than it might seem.

Results: line 129. The authors investigate the effect of Xyl phosphorylation on the catalytic efficiency of B3GALT6 and B3GAT3 but do not comment on whether Xyl phosphorylation affects the catalytic efficiency of B4GALT7, the enzyme transferring a Gal sugar to the Xyl in question. Please comment on how Xyl phosphorylation affects the catalytic efficiency of B4GALT7.

Gulberti et al. (PMID 15522873) showed that B4GALT7 transfers Gal to xylosides only when they are not phosphorylated at position 2. This is now mentioned on page 5.

Results: line 131. Suggest to directly refer to changes in catalytic efficiency rather than “increased the reaction” in the sentence, “We found that phosphorylation dramatically increased the reaction catalysed...”.

This has been changed as suggested.

Fig. S3B: Graph for B3GALT6: Need to put phosphorylated and unphosphorylated substrate curves on different scales to be able to evaluate these data. Also, for the unphosphorylated substrates, accurate K_M values cannot be accurately calculated when the maximum substrate concentration tested is at or below the K_M . Data with higher substrate concentrations for these samples are needed for accurate K_M values.

Supplementary Figure 3b has been changed as suggested. In all tables reporting kinetic constants, we now indicate values where the highest substrate concentration exceeds K_M . The error range of the fitted parameters is a good indication of whether the derived parameters are accurate or not. In several cases, no fit was obtained at all because the initial rates increased linearly with substrate concentration. We deliberately present the full experimental data rather than just the derived kinetic constants, so that these issues can be appreciated by the reader. More accurate K_M values would not change any of the conclusions of our study.

Fig. S3C Please include a panel with the AlphaFold model of B3GALT6 coloured by pLDDT.

This has been done.

Results: line 136. Propose to change “could be rationalised by structural modelling...” to reflect that an AlphaFold model was used and the substrate binding pocket was inferred by homology with a structure of a related enzyme.

This has been rewritten as suggested.

Fig 2B: Accurate Michaelis-Menten constants cannot be calculated when the highest substrate concentration used is not even the concentration needed for half V_{max} . Either repeat assays with a more appropriate range of substrate concentrations or mention that the K_M cannot be accurately determined for these reactions.

In all tables reporting kinetic constants, we now indicate values where the highest substrate concentration exceeds K_M . The error ranges of the fitted parameters are a good indication of whether the derived parameters are accurate or not. In several cases, no fit was obtained at all because the initial rates increased linearly with substrate concentration. We deliberately present the full experimental data rather than just the derived kinetic constants, so that these issues can be appreciated by the reader. More accurate K_M values would not change any of the conclusions of our study.

Table S3: Please report unit cell angles, X-ray wavelength used, as well as average B-factors

for the protein, waters and ligands.

This has been done.

Fig. S7A, please include a panel with the CSGALNACT2 ColabFold model coloured by pLDDT.

This has been done.

Results: For the models discussed in paragraphs starting on line 211 and 229, the structural insight would be substantially improved by including actual docking experiments where molecular dynamics simulations are used rather than manually placing substrates based on structural alignments. This is especially relevant for the paragraph starting on line 229 where specific molecular interactions are discussed. Given how these models were created, we suggest avoiding the discussion of specific residue-substrate interactions.

We agree with the reviewer that the CSGALNACT2-acceptor substrate model is speculative at this stage and have removed any reference to specific residues in the revised manuscript (page 9).

However, we believe that the EXTL3-acceptor substrate model is reliable. First of all, it is based on a 1.6 Å-resolution crystal structure of the enzyme instead of an AlphaFold model. Furthermore, in the NMR study of Agrawal et al. (PMID 10362836), the 13 best conformers of the tetrasaccharide were extremely similar, indicating an essentially rigid structure. Therefore, there are no uncertainties when extending from the experimentally determined position of the terminal GlcA-Gal disaccharide in EXTL2, whose GlcNAc transferase site is very similar to that of EXTL3 (see Figure 4c of PMID 35676258). The position of the Xyl-2-O-phosphate group is also defined within narrow limits and the phosphate group will be “close to Arg907” in any of the possible rotamers. Where there is indeed uncertainty is in the Xyl-Ser torsion angle and in the conformation of the peptide backbone. This is why we only indicate the general direction of the peptide, but refrain from making any predictions about specific interactions.

We do not believe that molecular dynamics simulations would change any of the qualitative conclusions we draw from the enzyme-substrate models. To facilitate further analysis by others, we are depositing the coordinates of our models as “Source Data”.

Reviewer #3

Major points:

1. Throughout the manuscript, the authors make a point that GAG attachment occurs in unstructured or disordered regions. I am curious whether the structure of these peptides/proteins varies between core proteins, and whether this could factor into the ability to be an EXTL3 substrate. Similarly, does the addition of a GAG chain change the conformation of such a peptide to impact its ability to be acted upon by EXTL3 and thus add a further determinant to HS/CS fate.

To our knowledge, no experimental structure exists of any GAG attachment site. Unstructured sites are flexible by definition, so the degree of variation is not easy to quantify. As mentioned in our response to a point raised by reviewer 1, we believe that GAG attachment sites must be located in unstructured regions to allow access to the xylosyltransferases that initiate the modification. It is quite possible that addition of a GAG chain changes the conformation or dynamics of the attachment site (perhaps favouring addition of another chain to another site nearby), but investigating this question experimentally would be an entire study in itself.

2. In lines 177-179, the authors state that SDC4 and GPC1 glycopeptides containing multiple closely spaced Ser-Gly sequons were both good substrates for EXTL3. Further, as these multiple sequons are suggested to be determinants of HS preference, providing further evidence to confirm that all SG sites in both proteins are occupied would be welcomed. Further, I postulate whether the mutagenesis of one or more of these SG sites would impact the identity of attached GAG chains. The authors refer to other PG peptides as evidence of single vs multi-site EXTL3 substrates not enhancing EXTL3 activity, but the amino acid sequences are quite different.

Shworak et al. (PMID 7520439) demonstrated that all three sites in syndecan-4 are occupied with HS chains, but we are not aware of a similar study with glypican-1. We know that all Ser-Gly sequons are modified with linker tetrasaccharides in our peptides (MS data in Supplementary Table 2). There is no evidence for partially modified peptides in the HPLC profiles of Tetra(P)-SDC4 and Tetra(P)-GPC1 (Supplementary Figure 4). For SDC4, we also have evidence (Supplementary Figure 5a, Supplementary Table 2) that both linker tetrasaccharides can be further modified with GlcNAc (EXTL3) or GalNAc (CSGALNACT2). Thus, we have every reason to believe that multiple Ser-Gly sequons are fully modified in our system, and likely also in the parent proteins.

Regarding the second point (single vs multi-site EXTL3 substrates), we have carried out an informative experiment (new Supplementary Figure 6). Eliminating one of the two attachment sites in the SDC4 peptide did not turn it into a poor substrate for EXTL3, confirming that flanking sequences indeed confer specificity.

3. The authors allude to this one pot strategy as being a potential method for the in vitro generation of GAGs. Thus, I am curious about the degree of polymerization/chain length achieved in their experiments. This could be partially achieved by performing disaccharide analysis on the final glycopeptide or SEC analysis.

The degree of polymerisation achieved with EXT1/EXT2 is high. We have extended our description of the in-gel fluorescence experiments (page 10) and carried out an experiment in which we varied the amount of sugar donors. In the new Supplementary Figure 9f, it is possible to count individual sugar additions up to \approx dp20 from the bottom of the gel. After that, individual bands are no longer resolved, but the maximum of the product distribution continues to increase in size and we believe that in the rightmost lane, the maximum is close to the theoretical value of dp200 (\approx 40 kDa). In Supplementary Figure 10c, the major

GAG form of sGPC1 runs between 140 and 260 kDa, also suggesting a long chain, and there are even longer products that didn't enter the gel at all.

For CS backbones, please see our response to point 5a.

4. I would like to see evidence to support the author's claim that tetrasaccharide linkers had been added to all three Ser-Gly sites in their soluble GPC1 protein. MS has been used throughout the manuscript to support the author's claims of tetrasaccharide linker modification, performing such analyses on GPC1 would provide much-needed support to the claim that all three sites are modified.

We think that the pronounced shift in mobility in Figure 5a demonstrates addition of several linker tetrasaccharides to the sGPC1 protein, especially given that identical treatment of the GPC1 peptide results in full modification of all three sites (see our response to point 2). Addition of a single linker tetrasaccharide to decorin core protein did not cause a similarly large shift on SDS-PAGE (response to point 5, new Figure 5b and Supplementary Figure 10d). In the revised manuscript, we say "that linker tetrasaccharides had been added to some or all of the three Ser-Gly sequons in the core protein" (page 11) to indicate our uncertainty about the precise stoichiometry of the *in vitro* modification.

We did attempt to verify the modification of sGPC1 by mass spectrometry. LC-ESI-MS of unmodified sGPC1 gave a mass of 61,240 Da, consistent with the calculated mass of 56,986 Da of the protein component and the presence of potentially two N-linked glycans. Modification with the linker enzymes resulted in a mass increment of $\approx 2,000$ Da, which is close to the mass of three phosphorylated linker tetrasaccharides ($712 \text{ Da} \times 3 = 2,136 \text{ Da}$). However, the data quality of the top-down intact molecular mass spectra is not sufficient to allow an unambiguous assignment. This is compounded by the fact that the modified glycoprotein is very likely not a homogeneous species, resulting in splitting and lowering of the signal intensities. A definitive glycoproteomic analysis of the target peptide bearing the linker tetrasaccharides is a very challenging experiment, given the length, acidity, and complexity of this particular tryptic glycopeptide region (33 amino acid residues, isoelectric point 3.4, cysteine nitrosylation in cells – PMID 19479373). We feel that such an analysis is beyond the scope of this manuscript.

a. If the "default" GAG chain is CS, I wouldn't expect to see a stunted CS backbone as the authors state in line 303. I wonder if the migration of HS and CS-modified GPC1 is simply different by SDS-PAGE, as the band appears to migrate slightly further than HS-modified GPC1 (further, in Fig. 3F and 3G, CS modified BETA migrated further than HS). A possible control for this would be native GPC1 enzymatically digested with heparinase and/or chondroitinase.

As is evident from Figures 4f and 4g, the CS polymerase we use (CHSY3/CHPF2) produces much shorter chains than the HS polymerase, EXT1/EXT2. The short chains polymerised by CHSY3/CHPF agree with data in PMID 17253960. The spacing of product bands is similar for HS and CS backbones, so we do not think that there is an inherent difference in electrophoretic mobility. We have carried out a new experiment with decorin core protein

(see point 5 below) and the CS backbone after overnight incubation is much more apparent than with sGPC1.

b. The GPC1 protein sequence possesses 6 Ser-Gly sites, I'm curious if the authors could postulate on why only three of these sites are typically modified?

The other Ser-Gly sites in human glypican-1 are Ser55, Ser308, and Ser372. In the crystal structure of the glypican-1 core protein (PDB code 4YWT), Ser55 is located at the end of a short β -strand, Ser308 is located in a tight reverse turn, and Ser372 is located at the start of an α -helix. None of these serines would fit into the active site of xylosyltransferase without disruption of the native glypican-1 fold.

5. The underlying discovery of the manuscript is the selective ability of EXTL3 to modify PG core proteins with HS chains instead of CS. As such, it would be nice to see the experiments performed in Figure 5 with a hybrid PG (such as Syndecan-1 or -3).

We have carried out the suggested experiment with decorin core protein, which *in vivo* contains a single CS/DS chain (new Figure 5b and Supplementary Figure 10d). The results not only confirm that HS cannot be attached to decorin, in agreement with our model, but they also show a more pronounced CS backbone smear than with sGPC1, possibly because the decorin core protein is smaller than sGPC1.

Minor points:

1. In lines 50-52 it is unclear what function the authors are specifically referring to, presumably the binding of growth factors?
“... in the nervous system, for instance, HS is an attractive signal, whereas CS is a repellent”

This process involves differential signalling via receptor protein tyrosine phosphatase σ . This information has been added on page 3.

2. The panels for Figure 4 are not referenced in order (A, D, E, B, C).

The section has been reorganised, so that the panels are now described in the correct order.

REVIEWERS' COMMENTS

Reviewer #2 (Remarks to the Author):

The authors have addressed our concerns / comments to our satisfaction. We recommend publication of the paper.

Reviewer #3 (Remarks to the Author):

The authors have satisfactorily addressed this Reviewer's concerns.